# InteractScience: Programmatic and Visually-Grounded Evaluation of Interactive Scientific Demonstration Code Generation

Qiaosheng Chen [1 2]   Yang Liu [1]   Lei Li [3]   Kai Chen [2]   Qipeng Guo [2]   Gong Cheng [1 †]   Fei Yuan [2 †]

## Abstract

While Large Language Models (LLMs) hold promise for automating science and education, generating interactive scientific demonstrations demands a complex synthesis of deep domain knowledge and precise reactive coding. Current benchmarks fail to capture this synergy, largely bifurcating into static code generation or text-only reasoning. To address this, we introduce INTERACTSCIENCE, the first benchmark dedicated to evaluating the holistic creation of interactive scientific applications. We propose a novel hybrid framework that integrates programmatic functional testing for logic verification with visually-grounded qualitative assessment for rendering fidelity. Our evaluation of 30 leading models across five disciplines reveals critical gaps in grounding scientific reasoning within interactive interfaces. By standardizing this combined capability, INTERACTSCIENCE establishes a crucial foundation for reliable AI-driven tools in science and education.

## 1. Introduction

Recent advancements in Large Language Models (LLMs) are catalyzing a fundamental shift in the paradigm of software creation, moving from a process of writing low-level code to one of articulating high-level, declarative goals (Comanici et al., 2025; OpenAI, 2025). Users now specify a desired outcome (such as "create a tool to visualize protein folding" or "build an interactive simulation of planetary orbits") and expect the LLM to translate this intent into a complete functional application (Chen et al., 2025; Sun et al., 2025c). This evolving human-AI collaboration is poised to

accelerate scientific research and education, empowering scientists to prototype data visualizations or educators to create bespoke interactive teaching modules, all articulated through natural language (Chu et al., 2025; Van Noorden & Perkel, 2023; Gottweis et al., 2025; Bai et al., 2025a; Sun et al., 2025d). Success is increasingly measured by how well the generated application produces a **functionally correct, visually intuitive, and interactive experience** that faithfully captures users' intended goals (Sun et al., 2024; Jiang et al., 2024).

In this new paradigm, we focus on **Scientific Demonstration Code Generation** (Ji et al., 2025). These demonstrations are not just static diagrams, they are interactive tools that bring abstract concepts to life, widely used in research and education for explaining complex scientific principles, supporting teaching, and presenting research findings. This task requires a model to translate abstract scientific principles into a tangible, interactive, and functionally correct system (Ji et al., 2025; Chen et al., 2025). However, this ambitious task exposes a critical limitation of current LLMs that we observed in practice. For example, as shown in Figure 1, a state-of-the-art LLM can easily explain Newton's second law or generate code for a blog webpage with standard UI elements. Yet, when asked to combine these skills to generate an interactive web demonstration of a block on an inclined plane, most models fail, producing errors ranging from incorrect physics logic in the JavaScript to non-functional UI components. *This highlights a fundamental gap that models can perform individual tasks but struggle to integrate them effectively* (Feng et al., 2025; Li et al., 2023).

At the same time, existing evaluation methodologies are insufficient for scientific demonstration code generation (Laskar et al., 2024; Chen et al., 2024). Current benchmarks either focus on knowledge question answering (Rein et al., 2024; Hendrycks et al., 2021) or static web code generation (Yun et al., 2024; Gui et al., 2025; Lu et al., 2025), but rarely assess the combination of functional interactivity and scientifically accurate visualization required in interactive demonstrations. Specifically, they lack reliable mechanisms to verify whether user actions correctly trigger the intended scientific behavior, relying on fixed-interval screenshots (Zhang et al., 2025a) or element-

[1] State Key Laboratory for Novel Software Technology, Nanjing University, Nanjing, China [2] Shanghai Artificial Intelligence Laboratory, Shanghai, China [3] Carnegie Mellon University, Pittsburgh, PA, USA. Correspondence to: Gong Cheng <gcheng@nju.edu.cn>, Fei Yuan <yuanfei@pjlab.org.cn>.

*Proceedings of the 43rd International Conference on Machine Learning*, Seoul, South Korea. PMLR 306, 2026. Copyright 2026 by the author(s).

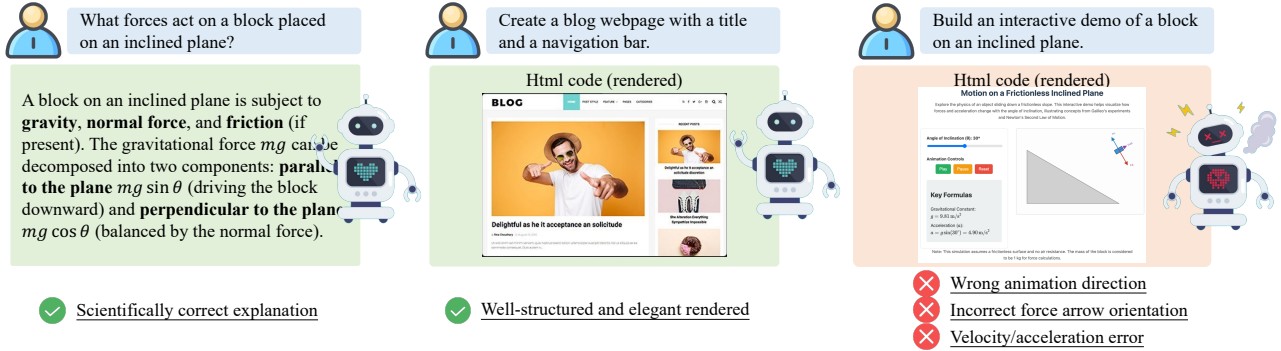

*Figure 1.* Illustration of three tasks. (a) **Knowledge Question Answering**: given the query about forces act of a block placed on an inclined plane, an LLM can output a correct textual explanation. (b) **Webpage Code Generation**: given the instruction of write a blog webpage, an LLM can generate functional static HTML code. (c) **Scientific Demonstration Code Generation**: generating an interactive demo for the inclined plane scenario, an LLM often fail to produce correct results.

existence checks (Xiao et al., 2025a) without actual interactions. Moreover, vision-based evaluations that use Vision-Language Models (VLMs) as judges (Gu et al., 2024; Li et al., 2024), typically without reference snapshots, tend to produce subjective judgments that fail to ensure scientific fidelity (Ji et al., 2025). These gaps make it difficult to measure whether a model successfully translates abstract scientific principles into a fully functional, interactive application.

To overcome these limitations, we design a hybrid evaluation framework that combines two complementary components. **Programmatic Functional Testing (PFT)** introduces deterministic unit test verification of interaction logic, ensuring that user actions trigger the intended behavior. **Visually-Grounded Qualitative Testing (VQT)** leverages target snapshots as visual oracles, providing grounded references for VLM-as-judge and enabling reliable assessment of visual correctness. Together, these two components form a robust methodology for evaluating scientific demonstration code. Building on this framework, we construct a new benchmark named **InteractScience**. It comprises a set of challenging problems across five diverse scientific disciplines: mathematics, physics, chemistry, earth science, and computer science. Each problem is accompanied by a complete evaluation suite, including unit test scripts for programmatic user behavior simulation and verification, reference snapshots for visually-grounded assessment of scientific correctness, and checklists for guidance of VLM-based semantic judgement. To probe the capabilities of current models, we conduct a large-scale evaluation of 30 prominent open- and closed-source models on InteractScience and provide an in-depth analysis of their performance. Our contributions can be summarized as follows:

1. We design a hybrid evaluation framework for scientific demonstration code generation, providing the first unified protocol that jointly validates interaction logic and visually grounded scientific fidelity.

2. We construct and release the InteractScience benchmark, firstly providing a rigorous evaluation suite with unit test scripts, reference snapshots, and checklist-guided judging.

3. We conduct extensive experiments across 30 state-of-the-art LLMs, providing a systematic analysis that exposes current limitations in integrating scientific reasoning with interactive code generation.

All code, data, evaluation outputs, and several complete examples are publicly available at https://github.com/open-compass/InteractScience.

# 2. Related Work

## 2.1. LLMs for Scientific Visualization.

Recent work has extended LLM evaluation to scientific and educational contexts. Benchmarks like SridBench (Chang et al., 2025) focus on generating scientific figures with semantic and structural accuracy, while EduVisBench (Ji et al., 2025) assesses pedagogically effective visual explanations for STEM education. These approaches emphasize domain knowledge but largely consider static visuals and do not evaluate interactive code or functional correctness. Other efforts treat interfaces as first-class outputs: Chen et al. (2025) demonstrate that LLMs can synthesize task-specific UIs with strong human preference, and TheoremExplainAgent (Ku et al., 2025) generates theorem explanations using Manim animations with automated metrics. Vinci-Coder (Zhao et al., 2025) uses reinforcement learning to enhance multimodal code generation, including scientific

*Table 1.* Comparison of InteractScience with related scientific visualization, software engineering, and code generation benchmarks. **SR**, **IT**, and **VT** denote Scientific Reasoning, Interaction-based Testing, and Visualization-based Testing, respectively.

| Benchmark | Primary Task | SR | IT | VT | Evaluation |
|---|---|---|---|---|---|
| *For Scientific Visualization:* | | | | | |
| SridBench | Scientific Visualization Generation | ✓ | ✗ | ✓ | LLM-judge |
| EduVisBench | Scientific Visualization Generation | ✓ | ✗ | ✓ | LLM-judge |
| *For Software Engineering:* | | | | | |
| SWE-bench | Repository-level Bug Fixing | ✗ | ✓ | ✗ | Automation |
| SWE-bench Multimodal | Repository-level Bug Fixing with Visual Context | ✗ | ✓ | ✗ | Automation |
| *For Code Generation:* | | | | | |
| LiveCodeBench | Competitive-Programming Code Generation | ✗ | ✗ | ✗ | Automation |
| ChartMimic | Chart Code Generation | ✗ | ✗ | ✓ | Automation + LLM-judge |
| MatPlotBench | Chart Code Generation | ✗ | ✗ | ✓ | LLM-judge |
| PandasPlotBench | Chart Code Generation | ✗ | ✗ | ✓ | LLM-judge |
| WebDev Arena | Web Design | ✗ | ✗ | ✓ | Human Voting |
| Interaction2Code | Interactive Web Page Generation | ✗ | ✓ | ✓ | Automation |
| WebGen-Bench | Interactive Web Page Generation | ✗ | ✗ | ✓ | LLM-judge |
| ArtifactsBench | Interactive Visual Artifacts | ✗ | ✗ | ✓ | LLM-judge |
| InteractScience | Interactive Scientific Demonstration Generation | ✓ | ✓ | ✓ | Automation + LLM-judge |

visualization, via iterative refinement with visual feedback. However, such evaluations focus on presentation quality or user perception rather than verifying event-driven correctness in executable, web-based scientific demonstrations. Due to the difficulty of assessing interactive behavior, most prior efforts still rely heavily on manual evaluation. *InteractScience fills this gap by providing an automated evaluation framework with faithful real-interaction simulation, jointly assessing visual quality and scientific correctness.*

## 2.2. Evaluation of Visual Code Generation.

Traditional code generation benchmarks such as HumanEval (Chen et al., 2021) and LiveCodeBench (Jain et al., 2025) focus on algorithmic logic via unit tests. These are followed by software engineering tasks like SWE-bench (Jimenez et al., 2024) and SWE-bench Multimodal (Yang et al., 2025c) that evaluate repository-level bug fixing but primarily focus on functional resolution rather than visual or scientific correctness. In parallel, visualization-oriented benchmarks such as PandasPlotBench (Galimzyanov et al., 2025), PlotCraft Zhang et al., 2025b, and ChartMimic (Yang et al., 2025b) target static graphics and chart reproduction. In the web domain, many prior works evaluate design-to-code generation primarily focusing on static layout fidelity rather than verifying interactive behavior (Yun et al., 2024; Laurençon et al., 2024; Gui et al., 2025; Si et al., 2025; Sun et al., 2025a; Xiao et al., 2025b; Awal et al., 2025). General UI benchmarks such as Interaction2Code (Xiao et al., 2025a) and ArtifactsBench (Zhang et al., 2025a), consider interactivity using screenshots or basic functional tests, but they often rely on heuristics or subjective VLM/LLM scoring, which can miss subtle event-driven bugs and limit reproducibility. While WebDev Arena (LMSYS Org, 2024) assesses web design

quality through human preference voting, it lacks scalability and deterministic verification. A systematic comparison of InteractScience against these existing benchmarks across multiple dimensions is presented in Table 1. *InteractScience fills this gap by unifying scientific expertise with functional interactivity, offering a rigorous automated framework that simultaneously validates scientific reasoning, interaction logic, and visual fidelity.*

## 3. Evaluation for Scientific Demonstration Code Generation

### 3.1. Task Definition

**Scientific Demonstration.** A Scientific Demonstration is an interactive web application with two coupled sections: a **control panel** containing UI elements (e.g., sliders, buttons, inputs) for parameter manipulation, and a **visualization canvas** (e.g., chart, animation, simulation) that dynamically renders the scientific concept. Its core lies in the interaction logic that binds controls to the canvas, ensuring that visual updates correctly reflect the underlying principles.

**Scientific Demonstration Code Generation.** In this work, the Scientific Demonstration Code Generation task is formalized as the creation of such code: given a detailed **implementation plan** $P$, which specifies page structure, HTML components, initial states and parameters, interaction logic, and visualization techniques, the goal is to generate a self-contained **front-end code** artifact $C$. This output is a single HTML file embedding CSS and JavaScript, which must render in a browser as a functional demonstration without external dependencies. By framing the task this way, we directly link the design specification $P$ to the resulting functional demonstration, highlighting the dual evaluation

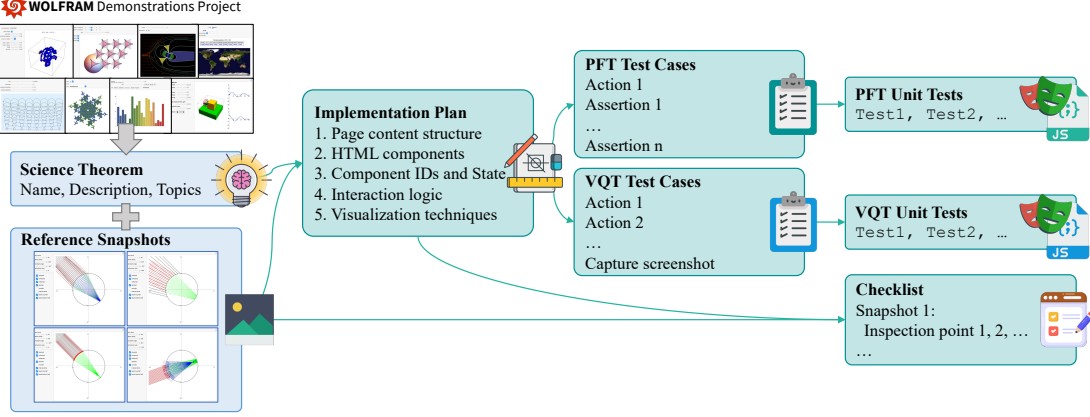

*Figure 2.* Pipeline of data collection and evaluation suite synthesis. The data collection step retrieves metadata of scientific demonstrations and corresponding snapshots from the Wolfram Demonstrations Project as seed data. The evaluation suite synthesis step generates implementation plans, test cases, unit tests, and checklist sequentially from the seed data.

requirements of code fidelity and scientific correctness.

## 3.2. Programmatic Functional Testing

Programmatic Functional Testing (PFT) provides deterministic verification of the component definitions in $P$, measuring whether the generated code $C$ behaves as specified.

**Formalism.** A PFT test case is an ordered sequence of action–assertion pairs

$$t_{\mathrm{pft}} = \{(a_i, e_i)\}_{i=1}^{N},$$

where each action $a_i$ is a simulated user interaction (e.g., a button click) and each assertion $e_i$ is a predicate on the expected DOM state. The complete test set $T_{\mathrm{pft}}(P)$ for a problem consists of as many unit tests $t_{\mathrm{pft}}$ as interactive components in $P$.

**Execution and Scoring.** An evaluation function

$$f_{\mathrm{pft}}(C, t_{\mathrm{pft}}) \to \{0, 1\}$$

executes the actions in $t_{\mathrm{pft}}$ on $C$. It returns 1 if all assertions $e_i$ hold, and 0 otherwise, providing an unambiguous measure of functional reliability.

## 3.3. Visually-Grounded Qualitative Testing (VQT)

Visually-Grounded Qualitative Testing (VQT) evaluates the correctness of the visualization and the visual quality of the generated demonstration, anchoring the assessment in explicit visual evidence.

**Formalism.** A VQT test case is a visual oracle

$$t_{\mathrm{vqt}} = (A, i_{\mathrm{ref}}, L),$$

where $A = (a_1, \ldots, a_k)$ is a sequence of user actions designed to reproduce the state depicted in the reference snapshot, $i_{\mathrm{ref}}$ is that corresponding reference snapshot, and $L = \{l_1, \ldots, l_m\}$ is a checklist of inspection points. The complete test set $T_{\mathrm{vqt}}(P)$ for a problem consists of as many unit tests $t_{\mathrm{vqt}}$ as reference snapshots provided for $P$.

**Execution and Scoring.** An evaluation function

$$f_{\mathrm{vqt}}(C, t_{\mathrm{vqt}}) \to (\text{CLIP Score, VLM-Judge Score})$$

executes the action sequence $A$ on the generated code $C$. The final action is to capture the screenshot, producing the generated snapshot $i_{\mathrm{gen}}$. The function then returns two complementary scores: **Perceptual Similarity**, computed as $\mathrm{CLIP}(i_{\mathrm{gen}}, i_{\mathrm{ref}})$ to measure low-level visual similarity, and **Semantic Correctness**, computed as $\mathrm{VLM}(i_{\mathrm{gen}}, i_{\mathrm{ref}}, L)$ to judge higher-level features guided by the checklist. These scores provide distinct perspectives on visual quality.

**Hybrid Evaluation Design.** The combination of PFT and VQT is not a simple aggregation of existing tools, because the two tracks are designed to expose different failure modes of interactive scientific code. PFT detects event-handling, state-update, and DOM-level functional failures that may not be visible from a single screenshot, whereas VQT captures scientific and visual errors that can pass DOM assertions but still render an incorrect phenomenon. In addition, checklist-guided judging converts open-ended visual assessment into fine-grained inspection criteria, making the VLM score substantially more reliable than either code-only or unguided visual judging.

*Table 2.* Statistics of InteractScience benchmark, where **Plan Len.** is the average length of plan, **#Cases** the average test cases, **#Act.** the average actions, **#Asrt.** the average assertions, and **#Check.** the average number of points in checklists, all computed per problem.

| Difficulty | #Prob. | Plan Len. | PFT | | | VQT | | |
|---|---|---|---|---|---|---|---|---|
| | | | #Cases | #Act. | #Asrt. | #Cases | #Act. | #Check. |
| Easy | 50 | 2055.84 | 2.54 | 5.68 | 11.36 | 3.98 | 6.44 | 21.56 |
| Medium | 50 | 2320.98 | 3.88 | 9.98 | 19.92 | 3.96 | 9.48 | 21.70 |
| Hard | 50 | 2586.34 | 5.68 | 15.40 | 30.64 | 4.00 | 13.74 | 23.02 |
| **Overall** | **150** | **2321.05** | **4.03** | **10.35** | **20.64** | **3.98** | **9.89** | **22.09** |

## 4. InteractScience Benchmark

### 4.1. Benchmark Composition

Each problem instance in the InteractScience benchmark is an evaluation suite containing three components: an implementation plan, a set of unit test scripts, and a set of evaluation checklists for the VLM-as-Judge.

**Implementation Plan.** Each benchmark problem is defined by a structured implementation plan with five parts: **(1) Page Content Structure.** Defines the logical UI sections (e.g., title, control panel, graph area, formula display) and their roles. **(2) HTML Components.** Lists HTML elements for each section (e.g., <div>, <canvas>) and notes libraries if needed. **(3) Component IDs and State.** Assigns each interactive element a unique ID and specifies default values, ranges, steps, and labels or tooltips. **(4) Interaction Logic.** Details how controls affect the application, including DOM updates, formula recalculations, visual rendering, and dependencies. **(5) Visualization Techniques.** Specifies rendering methods (e.g., D3.js, Plotly.js) and indicates which visuals require real-time updates or animations.

**Test Cases and Unit Test Scripts.** Derived from the implementation plan, each problem includes a suite of test cases to enable our hybrid evaluation framework. As described in Section 3, these are divided into two types. For PFT, we provide scripts composed of an alternating sequence of actions (e.g., simulating a button click) and assertions (e.g., verifying that a text value has updated correctly). For VQT, we provide separate, action-only scripts designed to reproduce the state shown in a target snapshot, culminating in a screenshot command. All test cases are provided as ready-to-run scripts in the Playwright [1] framework.

**Checklists for VLM-as-Judge.** As described in 3.3, each reference snapshot is paired with an evaluation checklist. This checklist is generated from the implementation plan and the underlying scientific principles of the task. It provides a rubric-based guide for the VLM judge, directing it to verify specific points of correctness. These points may in-

clude the accuracy of numerical values displayed, the proper alignment of graphical elements, the correctness of labels and legends, and the overall fidelity to the scientific concept being demonstrated. This anchors the VLM's assessment in the ground-truth specifications, moving beyond a purely open-ended visual interpretation.

### 4.2. Benchmark Curation

**Data Collection.** Our benchmark data is sourced from the *Wolfram Demonstrations Project*[2], which hosts over 13,000 interactive Wolfram Language notebooks. Unlike prior work relying on static materials like textbook (Ji et al., 2025), these executable demonstrations provide natural reference snapshots, ensuring reliable visual ground truth. From this corpus, we manually select 150 demonstrations across five disciplines, with the scale determined by the construction effort and the consideration of maintaining an acceptable evaluation cost (see Appendix A.2). To ensure diversity, we stratify by difficulty, defined by the number of interactive components: 1—3 (**easy**), 4—6 (**medium**), and 7—10 (**hard**), reflecting increasing interaction and visual complexity. For each demonstration, we collect metadata including title, description, topics, and associated snapshots.

**Evaluation Suite Synthesis.** As illustrated in Figure 2, starting from each demonstration's metadata and associated reference snapshots as seeds, we employ the state-of-the-art Gemini-2.5-Pro model to synthesize the corresponding implementation plans, test cases, unit test scripts, and evaluation checklists. The specific prompts used for synthesis are provided in the Appendix G. After each synthesis step, we apply manual inspection and rule-based validation to detect and correct obvious errors, ensuring that each test script is executable. In addition, we conduct a development experiment to verify the quality of the synthesized evaluation suite, as detailed in Section 5.3.

**Mitigating Synthesis Circularity.** Although parts of the evaluation suite are LLM-synthesized, the final scores are decoupled from model-generated text in three ways: PFT uses deterministic Playwright actions and binary DOM pred-

---

[1] https://playwright.dev

[2] https://demonstrations.wolfram.com/

icates with no LLM judge; VQT compares rendered screenshots against external Wolfram reference snapshots and checklist items; and all 150 problems undergo rule-based filtering and human review, followed by the quality checks in Section 5.3. Empirically, Gemini-2.5-Pro, the synthesis model, is not advantaged, as it ranks behind several GPT/Claude models on PFT OPR, and GPT-5 and Claude-Sonnet-4 also achieve higher VLM-Judge scores, suggesting that residual synthesis bias does not dominate the rankings.

## 4.3. Benchmark Statistics

The InteractScience benchmark includes 150 problems across five scientific disciplines (30 per discipline), with each discipline split into 10 easy, 10 medium, and 10 hard tasks (50 per difficulty overall). As shown in Table 2, difficulty correlates with complexity, with plan length and PFT rigor increasing from easy to hard (more test cases, actions, and assertions). Compared with prior work (Zhang et al., 2025a; Ji et al., 2025), our input plans are longer because they are structured design specifications rather than brief hints. The number of VQT test cases remains stable since each case corresponds to a reference snapshot, and nearly every problem includes four snapshots.

Although the benchmark contains 150 high-level tasks, each task is a complete interactive scientific web application and includes multiple executable checks. Overall, the benchmark contains 779 PFT unit tests and 590 VQT unit tests, yielding 1,369 individual tests. Across 30 evaluated models, this corresponds to 4,500 model–problem instances and more than 40,000 test executions. Thus, the benchmark is modest in problem count but dense in executable evaluation signals; continued expansion remains an important future direction.

## 5. Experiments

### 5.1. Experimental Setup

**Models.** We evaluate a broad range of state-of-the-art LLMs, including 10 closed-source and 20 open-source models. On the closed-source side, we include the **GPT** (Achiam et al., 2023; Hurst et al., 2024) series, the **Gemini-2.5** (Comanici et al., 2025) series, and the **Claude** series, which represent the most widely adopted commercial models. On the open-source side, we test **DeepSeek-V3** (Liu et al., 2024) and **DeepSeek-R1** (Guo et al., 2025), **Kimi-K2** (Team et al., 2025), **GLM-4.5** (Zeng et al., 2025), **Intern-S1** (Bai et al., 2025a), the **GPT-OSS** (Agarwal et al., 2025) series, the **Qwen3** (Yang et al., 2025a) series, the **Qwen2.5-Coder** (Hui et al., 2024) series, the **Qwen2.5-VL** (Bai et al., 2025b) series, and the **Llama-3.1** series.

**Metrics.** As described in Section 3, we evaluate models along two dimensions. For PFT, we report three pass-rate metrics: the **Overall Pass Rate (OPR)**, the percentage of unit tests passed across the entire benchmark; the **Average Pass Rate (APR)**, which averages pass rates over problems; and the **Perfect Pass Rate (PPR)**, the proportion of problems for which all unit tests pass. For VQT, we consider three aspects: the **Action Success Rate (ASR)**, the percentage of cases where the expected visual state appears after the specified action sequence; perceptual similarity measured by the **CLIP** score between the generated snapshot and the reference snapshot; and semantic correctness assessed by the **VLM-Judge** score (Gemini-2.5-Pro), which checks whether the visual result aligns with the task specification.

Evaluation details are presented in Appendix A.

## 5.2. Results

**Main Results on Scientific Demonstration Code Generation.** Table 3 summarizes the performance of all evaluated models on InteractScience. Despite differences across models, the absolute performance levels demonstrate that the benchmark is highly challenging. PFT scores remain modest, with PPR rarely exceeding 16%, underscoring the difficulty of generating code that flawlessly follows the implementation plan. On the visual side, ASR scores are consistently high, often above 85%. However, this metric primarily reflects surface-level interactivity; ASR is high because most models can reliably generate the specified UI components, allowing actions like "clicking a button" or "moving a slider" to execute without error, regardless of whether the resulting visualization is scientifically correct. This superficial competence does not transfer to deeper semantic fidelity. CLIP scores remain moderate and VLM-Judge scores are typically below 60, indicating that while models can create plausible-looking interfaces, they often fail to connect these designs with correct physical logic or scientific knowledge. *The gap between ASR and the semantic metrics thus reflects a key limitation, as models handle generic frontend generation well but struggle to integrate domain-specific reasoning into functional visualizations.*

Closed-source models generally outperform open-source models, especially in functional correctness, with Claude-Sonnet-4 and GPT-5 achieving the highest PFT scores, indicating stronger adherence to complex implementation plans. Among open-source models, larger ones such as DeepSeek-R1-0528 and Qwen3-235B-A22B-Instruct-2507 reach PFT scores comparable to mid-tier proprietary models, while smaller models ($\leq$14B parameters) struggle on both functional and visual tasks, highlighting the importance of model scale for this synthesis-heavy task. *Notably, these overall trends broadly align with human-annotated web development rankings such as WebDev Arena (LMSYS Org, 2024),*

*Table 3.* Main results of 10 closed-source and 20 open-source models on the InteractScience benchmark. CLIP and VLM-Judge scores are normalized to a 0–100 scale.

| Model | PFT | | | VQT | | |
|---|---|---|---|---|---|---|
| | Overall (%) | Average (%) | Perfect (%) | VLM-Judge | CLIP | Action (%) |
| *Closed-Source Large Language Models* | | | | | | |
| GPT-5 | 39.47 | **37.61** | **16.08** | **57.02** | 71.95 | 89.66 |
| GPT-4.1 | 37.07 | 34.08 | 11.19 | 52.84 | 71.21 | 89.15 |
| GPT-4o | 28.27 | 27.09 | 5.59 | 42.45 | 67.11 | 85.93 |
| o3 | 34.93 | 32.09 | 13.99 | 52.82 | 72.24 | 89.83 |
| o4-mini | 37.33 | 34.90 | 13.29 | 51.90 | 71.79 | 88.64 |
| Gemini-2.5-Pro | 35.33 | 34.62 | 11.19 | 54.69 | 70.65 | 86.78 |
| Gemini-2.5-Flash | 31.60 | 31.07 | 10.49 | 49.34 | 69.59 | 86.95 |
| Claude-Sonnet-4-20250514 | **41.47** | 37.40 | 13.29 | 55.42 | 73.50 | 89.66 |
| Claude-Opus-4-20250514 | 40.27 | 36.34 | 11.19 | 54.93 | **73.22** | 89.32 |
| Claude-3.5-Sonnet | 33.33 | 31.45 | 9.79 | 49.43 | 72.32 | **90.17** |
| *Open-Source Large Language Models* | | | | | | |
| DeepSeek-R1-0528 | **33.87** | **32.02** | 8.39 | 49.46 | 69.54 | 88.31 |
| DeepSeek-V3-0324 | 31.73 | 30.57 | 10.49 | 49.46 | 68.68 | 85.93 |
| Kimi-K2 | 31.60 | 31.22 | 9.79 | **50.04** | 70.11 | 87.29 |
| GLM-4.5 | 29.33 | 26.65 | 8.39 | 38.57 | 55.90 | 70.51 |
| Intern-S1 | 31.87 | 28.93 | 7.69 | 45.27 | 68.74 | 87.46 |
| gpt-oss-120b | 28.00 | 27.78 | 9.79 | 49.57 | **72.13** | **90.85** |
| gpt-oss-20b | 15.20 | 12.97 | 3.50 | 21.40 | 54.68 | 80.51 |
| Qwen3-235B-A22B-Instruct-2507 | 33.33 | 31.46 | **13.29** | 45.14 | 70.02 | 78.14 |
| Qwen3-32B | 27.20 | 24.09 | 5.59 | 39.69 | 66.46 | 87.46 |
| Qwen3-14B | 24.13 | 23.58 | 7.69 | 36.53 | 66.46 | 85.08 |
| Qwen3-8B | 20.00 | 18.85 | 4.20 | 34.67 | 64.13 | 81.53 |
| Qwen3-4B | 14.67 | 13.10 | 2.80 | 28.33 | 60.90 | 82.03 |
| Qwen3-1.7B | 6.53 | 6.22 | 1.40 | 20.33 | 59.65 | 75.76 |
| Qwen2.5-Coder-32B-Instruct | 27.20 | 25.10 | 7.69 | 38.51 | 51.67 | 84.58 |
| Qwen2.5-Coder-14B-Instruct | 22.53 | 20.61 | 4.90 | 35.72 | 64.47 | 85.42 |
| Qwen2.5-Coder-7B-Instruct | 12.40 | 10.51 | 0.70 | 26.97 | 65.17 | 82.37 |
| Qwen2.5-VL-72B-Instruct | 23.73 | 22.82 | 6.99 | 37.30 | 64.33 | 87.12 |
| Qwen2.5-VL-7B-Instruct | 7.47 | 6.72 | 0.70 | 20.41 | 49.49 | 70.00 |
| Llama-3.1-70B-Instruct | 18.67 | 18.04 | 4.90 | 33.36 | 59.56 | 88.64 |
| Llama-3.1-8B-Instruct | 11.33 | 10.16 | 3.50 | 22.75 | 65.42 | 80.00 |

*lending support to the reliability of our benchmark.*

Based on these findings, we recommend specific metrics for each evaluation dimension. For PFT, **OPR** is the preferred metric, as it reflects a model's ability to follow functional instructions by measuring adherence to all logical assertions in the plan. For VQT, **VLM-Judge** is the most informative indicator, as it directly captures the semantic and scientific correctness of the visualization. **CLIP** remains a useful lightweight proxy for perceptual similarity.

While these aggregate results provide a global view of model capabilities, performance varies across domains and difficulty levels; see Appendix B. To make the main failure pattern explicit, we manually inspect 90 outputs from three representative models on 30 sampled problems and assign non-exclusive error labels. Logic/scientific errors are the most frequent category overall (53.33%), followed by numerical/parameter errors (47.78%), rendering failures/basic errors (42.22%), and layout errors (38.89%). This taxonomy

supports the ASR–VLM gap observed above, showing that many generated demos expose usable widgets and event handlers even when the resulting visual states still violate the intended scientific semantics. Additional examples and per-model breakdowns are provided in Appendix D.

**Results on Multimodal LLMs with Reference Snapshots as Input.** To evaluate the impact of reference visual context, we test several multimodal LLMs by providing them with varying numbers of reference snapshots as part of the input. The models include GPT-5, GPT-4o, Gemini-2.5-Pro, and Qwen2.5-VL-72B-Instruct.

As shown in Figure 3, adding reference snapshots generally provides modest, model-dependent improvements, with GPT-5's PFT increasing from 39.47% to 42.53% and CLIP improving from 71.95 to 73.51, while Qwen2.5-VL-72B-Instruct occasionally degrades. These results suggest that snapshots help models capture visual style and layout, but the harder bottleneck is grounding the depicted state in the

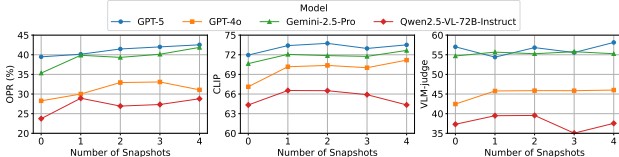

*Figure 3.* Performance of multimodal LLMs under varying numbers of reference snapshot inputs.

correct scientific or algorithmic semantics. A concrete case study is provided in Appendix E.

*Table 4.* Spearman correlations between PFT and VQT metrics.

| Model | PFT vs. CLIP | PFT vs. VLM-Judge |
|---|---|---|
| Gemini-2.5-Pro | 0.023 | 0.057 |
| GPT-5 | 0.081 | 0.155 |
| DeepSeek-V3 | 0.013 | 0.198 |
| Qwen3-8B | 0.158 | 0.312 |

To quantify the complementarity of the two tracks, Table 4 reports per-problem Spearman correlations between PFT and VQT metrics on 143 problems for four models. All correlations are low ($\rho < 0.32$), confirming that functional correctness and visual/scientific fidelity are not interchangeable. The weakest model, Qwen3-8B, has the highest correlation, suggesting that global failure can depress both tracks simultaneously, whereas stronger models often pass one dimension while failing the other.

*Table 5.* Spearman correlation between VLM-Judge scores and human expert scores under different input configurations.

| Judge Input | Corr. | Corr. p-value |
|---|---|---|
| $I_{\text{gen}} + I_{\text{ref}} + L$ | **0.8827** | $1.23 \times 10^{-26}$ |
| $I_{\text{gen}} + L$ | 0.8224 | $5.67 \times 10^{-23}$ |
| $I_{\text{gen}} + I_{\text{ref}}$ | 0.3837 | $1.01 \times 10^{-19}$ |
| $I_{\text{gen}}$ | 0.7360 | $9.87 \times 10^{-27}$ |
| $C$ | 0.1408 | $3.21 \times 10^{-6}$ |

**Comparison of VLM-as-Judge Configurations.** To validate our VLM-as-judge design, we measure how well different input configurations align with human judgment. Human experts score 30 randomly sampled outputs to establish ground truth, and we then conduct an ablation study by evaluating the same outputs with configurations that remove the reference snapshot ($I_{\text{ref}}$) or the checklist ($L$). We employ Spearman correlation between each configuration's scores and the human scores assesses alignment. The results in Table 5 show that the full configuration achieves the strongest alignment, highlighting the importance of both reference and checklist. Removing the checklist drops correlation to 0.3837, suggesting that without explicit checkpoints the VLM relies on coarse visual similarity, favoring outputs that appear plausible but fail scientifically. Judging with

only the textual code ($C$) yields negligible correlation, confirming that visual input is essential. *Together, reference snapshots and checklist-guided judging enable a more faithful and reproducible evaluation of scientific visualizations than unguided similarity-based scoring.*

*Table 6.* Pairwise Spearman correlations between VLM judge rankings.

| Judge Pair | Spearman $\rho$ |
|---|---|
| Gemini-2.5-Pro vs. GPT-5 | 0.937 |
| Gemini-2.5-Pro vs. Kimi-K2.5 | 0.919 |
| GPT-5 vs. Kimi-K2.5 | 0.956 |

We further test judge robustness by evaluating the same VQT rendered snapshots with GPT-5 and Kimi-K2.5 as alternative judges. As summarized in Table 6, the resulting rankings are highly consistent across judges ($\rho > 0.91$), and the Top-5 model ordering is preserved across judge choices. Detailed score levels and per-discipline reliability analyses are provided in Appendix C.

**Complementarity of CLIP and VLM-Judge.** Figure 4 illustrates how CLIP and VLM-Judge provide complementary perspectives on visual evaluation. In Figure 4a, a generated *Huffman Tree Encoding* demonstration receives a CLIP score of 75.54 and a VLM-Judge score of 97.14, showing that both visual similarity and semantic correctness are well preserved. By contrast, Figure 4b presents a generated *Spring-Mass-Damper System* demonstration with a high CLIP score of 82.74 but a low VLM-Judge score of 32.00. While the overall layout and graphical style are maintained, the 3D spring is rendered incorrectly, resulting in a clear scientific error. These examples show that perceptual similarity alone cannot guarantee correctness, and VLM judges are crucial for identifying semantic inaccuracies. Additional snapshots from different models are included in Appendix F. *Overall, CLIP provides a lightweight proxy for perceptual similarity, while VLM-Judge offers a more fine-grained and accurate assessment of semantic and scientific correctness*

### 5.3. Quality Verification of Synthesized Evaluation Suite

**Manual Validation Experiment.** To verify the quality of our synthesized evaluation suite, we conduct a manual validation on 30 questions (20% of the dataset), sampling one question per subject and difficulty level. Five graduate students rate each component on a 1–5 scale (higher is better) along two axes: **faithfulness** to the original demonstration and **correctness** of the content.

The validation follows a single-annotator-plus-reviewer protocol rather than a fully crossed multi-rater design. Each sampled problem is first inspected by one annotator and then independently reviewed by a second annotator, with

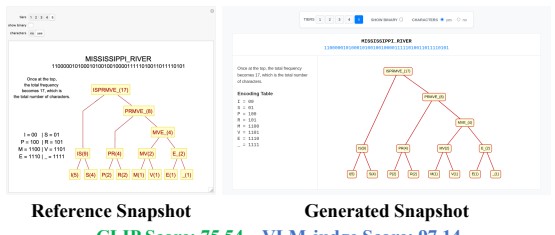

**Reference Snapshot**      **Generated Snapshot**
CLIP Score: 75.54     VLM-judge Score: 97.14

*(a)* A Huffman Tree Encoding demonstration.

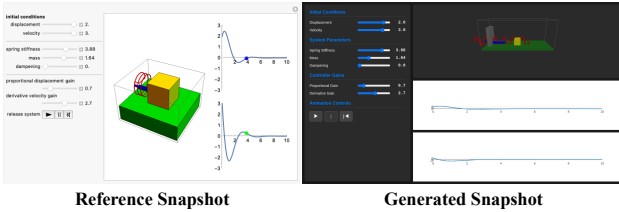

**Reference Snapshot**      **Generated Snapshot**
CLIP Score: 82.74     VLM-judge Score: 32.00

*(b)* A Spring-Mass-Damper System demonstration.

*Figure 4.* Examples comparing CLIP and VLM-Judge scores.

disagreements discussed and arbitrated before final scoring. Therefore, traditional multi-rater agreement statistics such as Krippendorff's alpha are not directly applicable. To support informed evaluation, annotators are provided with reference materials including the scientific theorem or concept explanation, demonstration usage guide, and expected parameter behaviors before annotation.

This validation differs from the benchmark construction stage. During construction, we perform an executable validity check on all 150 problems (rule-based filtering for syntax errors plus human review to fix execution issues such as DOM selectors), whereas the validation here targets the logic correctness of the synthesized suite.

*Table 7.* Manual rating results on sampled instances (Scale 1–5).

| Component | Faithfulness | Correctness |
|---|---|---|
| Implementation Plans | 4.43 | 4.73 |
| PFT Test Cases | 4.37 | 4.63 |
| VQT Test Cases | 4.43 | 4.67 |
| PFT Unit Test Scripts | 4.53 | 4.27 |
| VQT Unit Test Scripts | 4.57 | 4.23 |
| Checklists | 4.87 | 4.63 |

As shown in Table 7, all components score above 4 on average. Checklists are highest, while unit test scripts are slightly lower due to the inherent difficulty of producing executable test code. *These results provide strong evidence that our synthesized suite is both faithful and semantically reliable, supporting our subsequent automated evaluation.*

*Table 8.* Correctness of unit test execution across different models.

| Model | PFT(%) | VQT(%) |
|---|---|---|
| Gemini-2.5-Pro | 87.5 | 91.2 |
| DeepSeek-V3 | 86.8 | 92.4 |
| Qwen3-8B | 95.3 | 96.7 |

**Visual Verification of Test Execution.** Recognizing that unit test scripts are difficult to validate by reading code alone, we utilize Playwright UI Mode [3] to verify the test execution logic. For 30 questions, we execute the scripts on the code generated by three models (Gemini-2.5-Pro, DeepSeek-V3, and Qwen3-8B), resulting in 90 unit test executions. We judge whether each script produces the correct test behavior (i.e., avoiding false positives/negatives).

As shown in Table 8, the correctness rates consistently exceed 86%. VQT scripts show higher correctness as they generally involve fewer logic assertions. Notably, for lower-capability models (Qwen3-8B), script correctness is higher because the generated code is simpler, making the tests behave more predictably. *These results indicate that the unit test executions are reliable, providing confidence that our automated evaluation accurately reflects model behavior.*

## 6. Conclusion

In this work, we address the challenge of evaluating the ability of LLMs to integrate scientific knowledge into interactive demonstrations. We introduce InteractScience, the first benchmark for scientific demonstration code generation with a hybrid evaluation framework that combines deterministic PFT for verifying interactive functionality and reliable VQT for assessing visual fidelity. By targeting executable scientific demos rather than static answers, InteractScience offers a practical testbed for AI-assisted science education.

While our study demonstrates the feasibility of evaluating scientific demonstration code generation, the current dataset is limited in data size and expert verification, which may leave some subtle interactive or semantic cases untested. These aspects suggest opportunities for future work, including expanding the benchmark, incorporating broader expert validation, and exploring agent-driven testing to improve coverage, flexibility, and scalability. More broadly, InteractScience can provide diagnostic signals for improving models' scientific reasoning, visual grounding, and interactive programming abilities. A more detailed discussion is provided in Appendix H. We believe that our work will contribute to more reliable AI tools for science and education applications.

---

[3] https://playwright.dev/docs/test-ui-mode

## Acknowledgements

This work is supported by the Fundamental Research Funds for the Central Universities (No. 2026300429), the Fundamental and Interdisciplinary Disciplines Breakthrough Plan of the Ministry of Education of China (No. JYB2025XDXM118), and the "111 Center" (No. B26023). We thank Qiushi Sun and Jingyang Gong for their helpful suggestions and discussions on improving this work.

## Impact Statement

This paper presents work whose goal is to advance the field of machine learning. The benchmark is constructed from public resources and does not involve any personal or sensitive information, and it is intended solely for research and educational purposes. We expect INTERACTSCIENCE to support more transparent and reproducible evaluation of LLM-generated web applications and scientific visualizations, potentially benefiting education and scientific communication.

At the same time, AI-generated scientific demonstrations may create societal risks if visually plausible but scientifically incorrect materials are used in low-supervision educational settings. Benchmark scores should therefore not be interpreted as certification of scientific correctness, nor should generated demonstrations be deployed to learners without expert review. By exposing failures in interactive scientific reasoning, INTERACTSCIENCE is intended as an auditing tool that helps identify such risks rather than a replacement for domain-expert validation.

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

# A. Evaluation Details

## A.1. Evaluation Environment

All testing after obtaining model outputs was conducted on a single server node equipped with 64 CPU cores and 512 GB of RAM. For the experiments, closed-source models and open-source models with more than 72B parameters were evaluated via standard API calls with default configuration. Open-source models with fewer than 72B parameters were deployed and run on a setup of 8 NVIDIA H800 GPUs, each with 80 GB of memory. During inference, the temperature was set to 0, and the maximum context length for open-source models was set to 32,000 tokens.

## A.2. Evaluation Cost

We report the computational and financial cost of evaluating InteractScience. The benchmark comprises 779 PFT unit tests and 590 VQT unit tests. Using 96 concurrent processes, the average runtime per problem is 4.15 minutes for PFT and 3.56 minutes for VQT. In worst cases, due to code errors triggering repeated timeouts, evaluation can take up to around 10 minutes. Semantic correctness in VQT is assessed using Gemini-2.5-Pro as the VLM-as-Judge. Each evaluation round involves 597 judgment queries, incurring an average cost of 8–10 USD. This demonstrates that our evaluation is practical and economically friendly, making the benchmark accessible for future research and reuse.

## A.3. VQT Score Scaling

For VQT, the VLM-as-Judge assigns a score on a 1–5 scale for each snapshot, reflecting the degree of semantic and scientific correctness. If the corresponding input state fails to execute all specified actions and thus produces no snapshot, the case is assigned a score of 0. Consequently, the effective scoring range becomes 0–5. For presentation clarity, we linearly rescale these scores to a 0–100 range in the reported tables and figures.

## A.4. Metrics Computation

We evaluate models using two primary sets of metrics: PFT for code execution logic and VQT for visual correctness. Both evaluations rely on **Playwright** automation scripts to determine the success of test cases. Let $N$ be the total number of problems in the benchmark.

**PFT Metrics.** For each problem $i$, let $T_i$ be the set of unit test scripts generated to verify functional requirements. Each script $t \in T_i$ contains a sequence of interaction actions and logic assertions. The execution result, denoted as $pass(t)$, is determined automatically by the Playwright engine. A unit test is considered passed ($pass(t) = 1$) **if and only if all specified actions are successfully executed and all assertions are verified**. If any action fails (e.g., element not found) or any assertion fails, $pass(t) = 0$.

- **Overall Pass Rate (OPR):** The micro-average pass rate, calculated as the ratio of passed unit tests to the total number of unit tests across the entire benchmark.

$$\text{OPR} = \frac{\sum_{i=1}^{N} \sum_{t \in T_i} pass(t)}{\sum_{i=1}^{N} |T_i|} \times 100\% \tag{1}$$

- **Average Pass Rate (APR):** The macro-average pass rate. We first calculate the pass rate $r_i$ for each problem $i$, then average these rates.

$$r_i = \frac{\sum_{t \in T_i} pass(t)}{|T_i|}, \quad \text{APR} = \frac{1}{N} \sum_{i=1}^{N} r_i \times 100\% \tag{2}$$

- **Perfect Pass Rate (PPR):** The proportion of problems where the model passes *all* associated unit tests.

$$\text{PPR} = \frac{1}{N} \sum_{i=1}^{N} \mathbb{I}(r_i = 1) \times 100\% \tag{3}$$

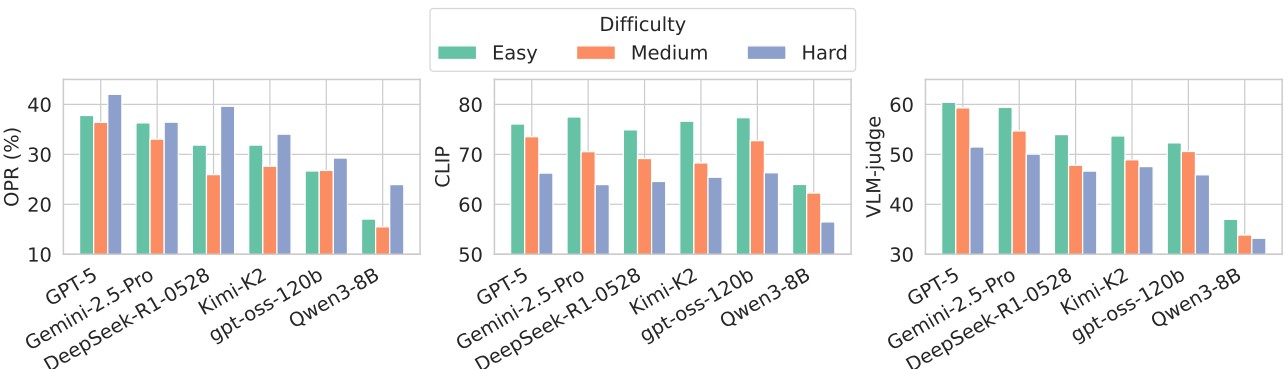

*Figure 5.* Performance of LLMs across different difficulty levels.

**VQT Metrics.** For each problem $i$, let $C_i$ be the set of visual test cases. Each case $c \in C_i$ consists of a Playwright script containing **only interaction actions** (without logic assertions) intended to reach a specific state for snapshotting.

- **CLIP Score:** If the action sequence executes successfully ($exec(c) = 1$), we capture a generated snapshot $S_c$. We compute the cosine similarity between $S_c$ and the reference snapshot $R_c$. If execution fails ($exec(c) = 0$), the snapshot is missing, and the score is 0.

$$\text{CLIP} = \frac{1}{\sum_{i=1}^{N} |C_i|} \sum_{i=1}^{N} \sum_{c \in C_i} \text{Sim}_{\text{CLIP}}(S_c, R_c) \tag{4}$$

- **VLM-Judge Score:** We employ Gemini-2.5-Pro to evaluate the semantic correctness of the generated snapshot $S_c$ against the task requirements, assigning a raw score $s_{raw}(c) \in [1, 5]$. If execution fails ($exec(c) = 0$), the score is 0. We linearly rescale the effective range $[0, 5]$ to $[0, 100]$.

$$v_c = \begin{cases} 0 & \text{if } exec(c) = 0 \\ s_{raw}(c) & \text{if } exec(c) = 1 \end{cases} \tag{5}$$

$$\text{VLM-Judge} = \frac{1}{\sum_{i=1}^{N} |C_i|} \sum_{i=1}^{N} \sum_{c \in C_i} \left( \frac{v_c}{5} \times 100 \right) \tag{6}$$

- **Action Success Rate (ASR):** This metric measures the stability of the generated application. Let $exec(c)$ be the execution status returned by Playwright. The test is considered successful ($exec(c) = 1$) if **all actions in the sequence are executed without error** (e.g., no timeouts or crashes).

$$\text{ASR} = \frac{\sum_{i=1}^{N} \sum_{c \in C_i} exec(c)}{\sum_{i=1}^{N} |C_i|} \times 100\% \tag{7}$$

## B. Performance Analysis Across Difficulty Levels and Disciplines.

Figure 5 shows performance across difficulty levels. We observe a counter-intuitive trend where the Overall Pass Rate (OPR) for PFT is higher on "Hard" problems compared to "Easy" ones. This phenomenon stems from the definition of difficulty (based on the number of interactive components) and the specific distribution of component types.

Our analysis reveals that "Hard" problems (7–10 components) often contain a high density of simple binary controls, such as checkboxes (e.g., 36% of components in Physics "Hard" tasks), which are syntactically easier to implement. In contrast, "Easy" problems (1–3 components) frequently rely on complex parameterized controls like sliders (e.g., requiring min/max/step attributes), which pose a greater functional implementation challenge. Consequently, models achieve higher functional compliance (PFT) on "Hard" tasks due to the simplicity of the individual code elements. However, simple code

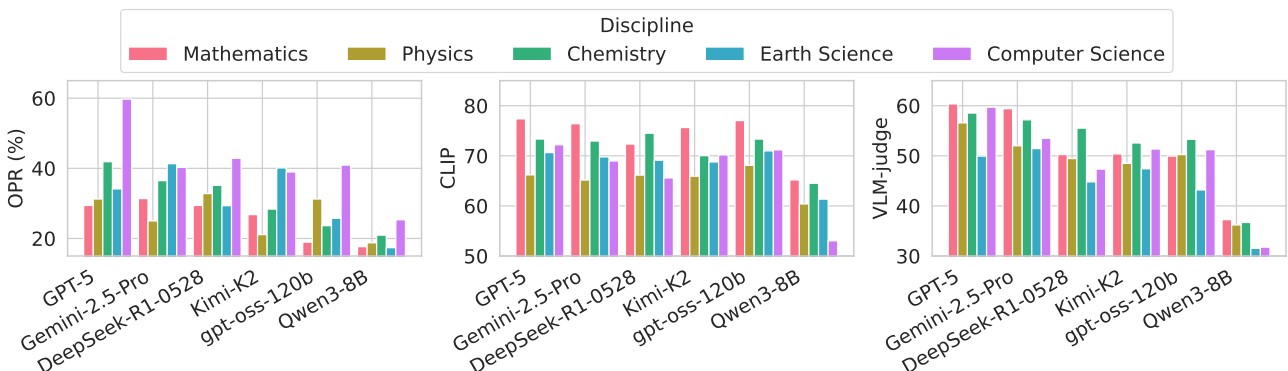

*Figure 6.* Performance of LLMs across different disciplines.

*Table 9.* Per-discipline Spearman correlations between VLM-Judge scores and human scores.

| Discipline | Spearman $\rho$ |
|---|---|
| Mathematics | 0.803 |
| Physics | 0.812 |
| Chemistry | 0.962 |
| Earth Science | 0.702 |
| Computer Science | 0.691 |
| Overall | 0.883 |

logic does not imply simple scientific interaction. The VQT scores decline steadily as difficulty increases, indicating that while models can generate the boilerplate code for numerous simple controls, they struggle to correctly render the complex visual dynamics and scientific effects associated with high-component-count tasks. *These results highlight a divergence that functional correctness (PFT) is influenced by the syntactic complexity of individual controls, while visual fidelity (VQT) is strongly negatively correlated with the overall complexity of the scientific task.*

This observation also reveals a limitation of our current difficulty calibration, since component count is an interpretable and reproducible proxy but does not fully capture interaction type diversity, formula complexity, or visualization dynamism. A more principled stratification could weight continuous controls more heavily than binary controls and incorporate scientific computation and rendering complexity. We leave this composite difficulty metric as future work, while retaining the current definition for transparency and ease of reproduction.

Figure 6 shows performance across disciplines. Models achieve higher PFT scores on disciplines like Computer Science and Chemistry. This is likely because these subjects frequently employ discrete, standard control interfaces (e.g., buttons for sorting algorithms or dropdowns for chemical elements), which are well-represented in training data and syntactically straightforward. In contrast, VQT scores drop significantly for Physics and Earth Science. These fields demand continuous dynamic simulations (e.g., projectile motion or planetary orbits) requiring complex frame-by-frame updates in the visualization canvas, where logic errors easily lead to visual artifacts. Conversely, Mathematics maintains high VQT scores, as many tasks involve static plotting or geometric transformations that rely on declarative rendering logic rather than complex physical simulation loops. *These patterns indicate that different disciplines place distinct demands on model capabilities, with some emphasizing accurate control logic and others requiring sophisticated visual rendering.*

## C. Judge Reliability and Robustness

To further examine judge reliability across domains, we compute Spearman correlations between VLM-Judge scores and human scores by discipline on the human-rated subset, as shown in Table 9. The lower correlations for Earth Science and Computer Science mainly arise from dense multi-panel visualizations and algorithmic layouts with several partially correct states, whereas Chemistry benefits from clearer structural criteria. These results indicate that the VLM-Judge is broadly aligned with humans but not uniformly reliable across all scientific domains.

Manual inspection of 200 VLM-Judge decisions further shows a misjudgment rate below 5%. Table 10 summarizes the

*Table 10.* Observed VLM-Judge failure modes from manual inspection of 200 decisions.

| Failure Mode | Description |
|---|---|
| Numerical misreading | Misreading axis values or parameter labels. |
| Stylistic confusion | Confusing color or thickness differences with scientific errors. |
| Partial-credit blindness | Overlooking partial correctness in dense screenshots. |
| Layout sensitivity | Penalizing valid alternative layouts. |
| Animation snapshot | Misjudging dynamic content from static captures. |

*Table 11.* VLM-Judge scores from three judge models on four evaluated model outputs.

| Judge | Gemini | GPT-5 | DeepSeek | Qwen3 | Avg. |
|---|---|---|---|---|---|
| Gemini-2.5-Pro | 54.7 | 57.0 | 49.5 | 34.7 | 49.0 |
| GPT-5 | 57.3 | 59.7 | 51.4 | 36.6 | 51.3 |
| Kimi-K2.5 | 50.8 | 52.8 | 46.6 | 33.3 | 45.9 |

main judge failure modes. Our checklist-guided design mitigates these issues by decomposing visual judgment into concrete inspection points, but temporal and highly dense demonstrations remain challenging.

**Judge Robustness.** To test whether benchmark outcomes depend heavily on Gemini-2.5-Pro as the judge, we additionally evaluate the same VQT rendered snapshots from four representative generation models using GPT-5 and Kimi-K2.5 as alternative VLM judges. As shown in Table 11, the absolute score levels vary moderately across judges, but the relative ordering of evaluated models is stable. Together with the main-text rank-correlation analysis, these results suggest that reference snapshots and structured checklists provide a robust judging protocol across different VLM backbones.

## D. Failure Analysis

**The "Superficial Competence" Gap.** In Section 5.2, we observe a dominant failure pattern where models often achieve high Action Success Rates (ASR; frequently $> 85\%$) but substantially lower VLM-Judge scores (often $< 55$). This suggests that models can correctly implement the "shell" of interactive demos (i.e., UI widgets and event handlers that execute), yet fail to bind controls to the underlying scientific logic, leading to visually plausible but scientifically incorrect behavior.

**Illustrative example.** Figure 4b shows a *Spring-Mass-Damper System* output with a high CLIP score but a low VLM-Judge score, where the overall rendering style appears convincing while the 3D spring is rendered incorrectly, producing a clear scientific error.

**Error taxonomy from manual inspection.** To ground this gap with qualitative evidence, we randomly sample 30 problems and manually inspect outputs from three representative models (Gemini-2.5-Pro, DeepSeek-V3, and Qwen3-8B), resulting in 90 inspected cases. We label errors with four non-exclusive categories (a case may contain multiple error types): (1) Rendering Failure/Basic Error, (2) Logic/Scientific Error, (3) Layout Error, and (4) Numerical/Parameter Error; the aggregated statistics are reported in Table 12.

**Analysis Results and Findings.** As summarized in Table 12, **Logic/Scientific Errors** are the most prevalent failure mode overall, consistent with the observed ASR–VLM gap. Models frequently produce interactions that execute reliably yet still violate the intended scientific semantics. Moreover, smaller models (e.g., Qwen3-8B) exhibit a markedly higher rate of **Rendering Failures/Basic Errors**, which can undermine visual evaluation even when UI actions complete successfully.

## E. Multimodal Reference Snapshot Case Study

The modest improvement from adding reference snapshots does not mean that multimodal models ignore visual inputs. We manually inspect a representative *Simple Caesar Cipher* demonstration to understand the effect of visual conditioning. As shown in Figure 7, without reference images, GPT-4o produces an interface with blue keyboard keys and symbolic controls for space/backspace. When the reference snapshot is provided, the model changes these elements to grey keys and text-label space/backspace controls, matching the reference style more closely. This qualitative difference indicates that the model

*Table 12.* Failure analysis on 30 sampled problems (90 inspected outputs across three models). Error labels are non-exclusive; percentages are computed over 30 outputs per model and over 90 outputs overall.

| Error Category | Gemini-2.5-Pro | DeepSeek-V3 | Qwen3-8B | Overall |
|---|---|---|---|---|
| Rendering Failure/Basic Error | 6 (20.00%) | 6 (20.00%) | 26 (86.70%) | 38 (42.22%) |
| Logic/Scientific Error | 18 (60.00%) | 20 (66.70%) | 10 (33.30%) | 48 (53.33%) |
| Layout Error | 14 (46.70%) | 16 (53.30%) | 5 (16.70%) | 35 (38.89%) |
| Numerical/Parameter Error | 10 (33.33%) | 13 (43.33%) | 20 (66.67%) | 52 (47.78%) |

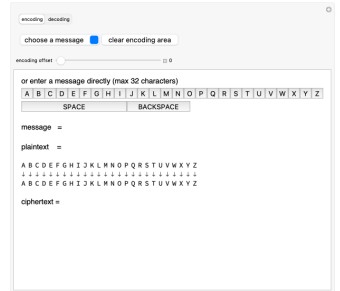
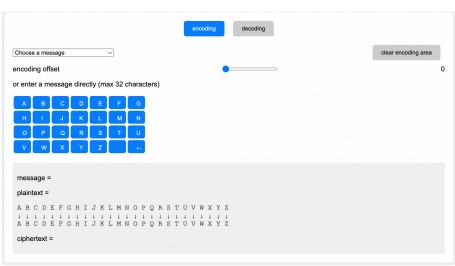
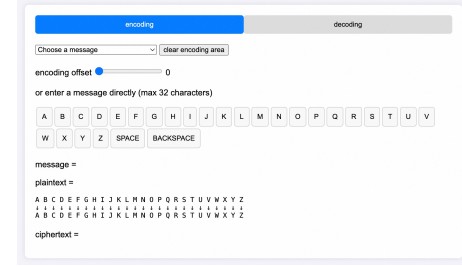

(a) Reference snapshot.  (b) Generated without snapshots.  (c) Generated with snapshots.

*Figure 7.* Case study on the *Simple Caesar Cipher* demonstration, comparing the reference snapshot with GPT-4o outputs generated without and with reference snapshots input.

does extract concrete visual details from the snapshot, especially color, layout, and control-label conventions.

However, the same example also illustrates why the aggregate gain remains small. Matching the visual style is only part of the task, since the model must also infer how the cipher state, keyboard controls, encoded text, and interaction logic should co-vary. Reference images provide evidence about the desired rendered state, but they do not directly specify the underlying algorithmic dependencies. As a result, additional snapshots can improve surface fidelity while simultaneously increasing the reasoning burden, especially when the model must infer scientific or algorithmic semantics from static visual evidence. This explains why reference snapshots yield modest average improvements and can occasionally hurt models that struggle to ground visual cues in correct interactive logic.

## F. Model Output Samples

Figures 8 and 9 present the reference snapshots alongside the outputs rendered by GPT-5, Gemini-2.5-Pro, DeepSeek-R1-0528, and Qwen3-8B for two benchmark demonstrations, *Fields of Magnet Array* and *Interwoven Spherical Triangles*. Each visualization is accompanied by its CLIP and VLM-Judge scores. These examples show that for complex demonstrations, different models exhibit varying levels of fidelity in both functional rendering and scientific visualization, with the quantitative scores reflecting the perceived visual quality and correctness.

## G. Prompts

We provide the full system prompts used to synthesize the evaluation suites of InteractScience. These prompts guide Gemini-2.5-Pro in generating implementation plans (Section G.1), PFT test cases (Section G.2), VQT test cases (Section G.3), PFT unit test scripts (Section G.4), VQT unit test scripts (Section G.5), VQT checklists (Section G.6), and VLM judgments (Section G.7), ensuring consistency and reproducibility of the benchmark.

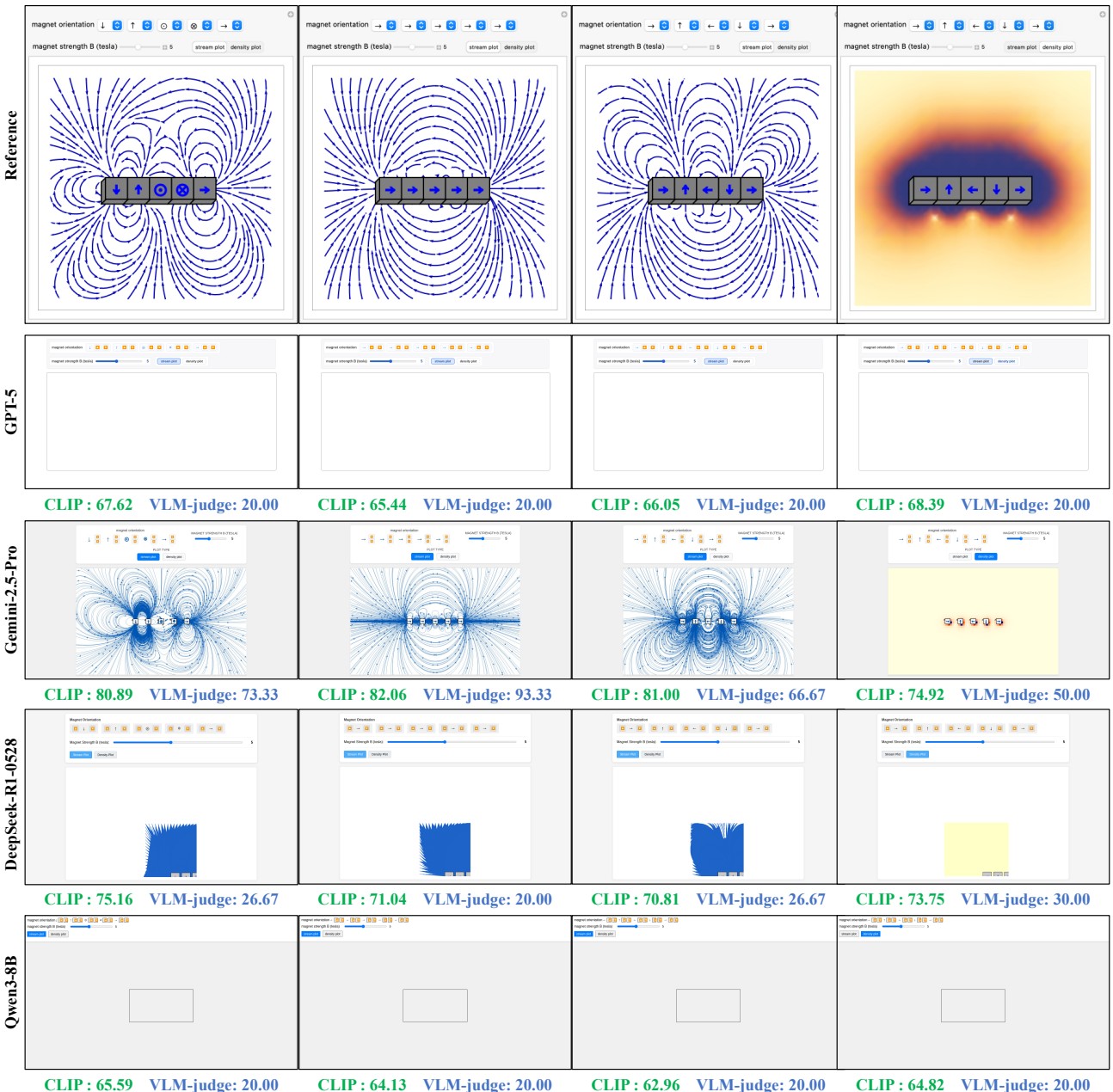

*Figure 8.* Reference and generated snapshots of different models for a Fields of Magnet Array demonstration.

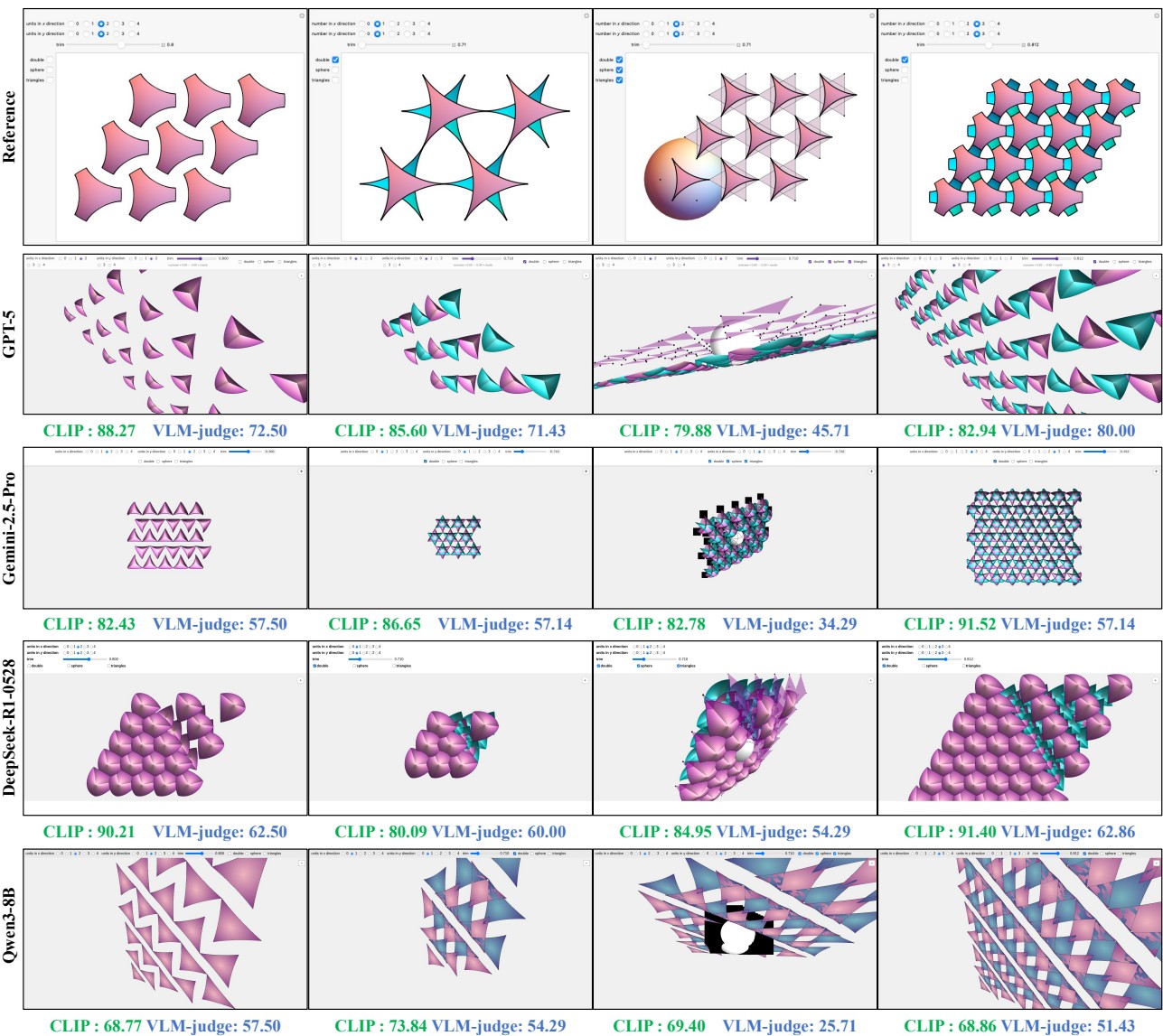

*Figure 9.* Reference and generated snapshots of different models for a Interwoven Spherical Triangles demonstration.

## G.1. Implementation Plan Synthesis

---

**System Prompt - Implementation Plan Synthesis**

You are an expert in frontend web development and scientific visualization. You are given the title, description, topics, and one or more screenshots of an interactive scientific demo. This demo is designed to explain a mathematical or scientific theorem or concept through visual interaction.

Your task is to generate a precise **implementation plan** for an interactive scientific demo that explains a specific mathematical or scientific theorem. Based on the input, produce a **complete, structured, and technically feasible implementation plan** using only standard web technologies.

This plan will be used as input for a large language model to reproduce the original demo. Therefore, you must provide **exhaustive specifications** for every component: layout, content, component IDs, initial values, interactions, and visual logic.

### Constraints:
1. The demo must be implemented as a single standalone HTML file with inline HTML, CSS, and JavaScript.
2. External libraries may only be included via **CDN** (e.g., p5.js, three.js, D3.js, Plotly.js, MathJax).
3. Do **not** reference any external or uploaded media assets (images, videos, audio), and do **not** use base64-encoded binaries.
4. The plan **must fully describe the observed UI state and behavior** in the provided screenshots (including default values, rendered formulas, slider settings, etc.).
5. Require the large language model that uses this plan to **strictly follow the implementation instructions**, since any missing information will lead to incorrect results.

### Output Format (strictly follow this structure, no extra commentary or code):
1. Page Content Structure
Describe each logical UI section (e.g., Title, Description, Control Panel, Graph Area, Formula Display) and its role.
2. HTML Components
List **all required HTML elements**, grouped by section. Include their types (e.g., `<div>`, `<input type="range">`, `<canvas>`, `<button>`, `<select>`).
Note if MathJax is required for formula rendering.
3. Component IDs and State
For every interactive component (sliders, checkboxes, dropdowns, buttons, etc.):
- Assign a unique `id` (e.g., `slider-angle`, `btn-play`)
- Provide: Initial/default value; Minimum and maximum (and step, if applicable); Label text or tooltip, if any
4. Interaction Logic
Explain **exactly** how each control affects the interface. For each user interaction, describe:
- What changes in the visual (e.g., redraws, updates)
- What dependent values or formulas update
- Whether animation or resets are triggered
Do not omit any interaction shown in the screenshots.
5. Visualization Techniques
Specify the rendering strategy and technology for each visual element:
- p5.js or Canvas API for custom 2D graphics
- three.js for 3D scenes
- D3.js or SVG for dynamic diagrams
- Plotly.js for charts or plots
- leaflet.js for maps
- MathJax for math formula rendering
- CSS for styling and layout (e.g., flex/grid, transitions, color indicators)
Indicate which elements require real-time updates or animation.

---

The resulting plan must be detailed enough that a large language model can accurately reproduce the entire original demo, including all interactions and visuals.

Here is the imformation of this scientific demo:
Name: { Name }
Description: { Description }
Topics: { Topics }
Snapshots: { Snapshots }

## G.2. PFT Test Case Synthesis

**System Prompt - PFT Test Case Synthesis**

You are an expert in frontend web development and scientific visualization. You are given the title, description, topics, one or more screenshots, and a detailed HTML implementation plan of an interactive scientific demo. Your task is to generate **component-level test cases** for each interactive element described in the plan.

Each test case should correspond to **exactly one component**, such as a slider, button, checkbox, dropdown, or any user-manipulable control.

### Test Case Requirements:
- The test case must validate:
1. The component is visible on page load.
2. The component has the correct default value or state (as defined in the plan).
3. The component can be interacted with correctly (e.g., drag slider, click button).
4. Boundary behavior should be tested (e.g., min/max values, reset, toggle on/off).
5. The interaction causes some change to the diagram, equation, UI element, or output (verify change occurred, not correctness).

### Output Format:
For each component, write one test case in the following format:
- Title: [Short description of the control being tested]
- Steps & Assertions:
1. Assert: [Component is visible]
2. Assert: [Component has correct default value or state]
3. Action: [Perform a realistic user interaction]
4. Assert: [UI update or state change occurred]
5. Action: [Boundary interaction or reset]
6. Assert: [System handles boundary or reset with some change]

### Guidelines:
- The difficulty of each test case should be **moderate**—not overly simple, not overly complex.
- Do **not** invent behavior not described in the implementation plan.
- Use only what is described or visible in the plan and screenshots.
- One case per component.
- Focus on detecting **change** rather than validating scientific correctness.
- Keep your language concise and precise—do not add explanations or commentary.

Here is the imformation of this scientific demo and the corresponding implementation plan:
Name: { Name }
Description: { Description }
Topics: { Topics }

Implementation plan:{Implementation plan}
Snapshots: { Snapshots }

## G.3. VQT Test Case Synthesis

**System Prompt - VQT Test Case Synthesis**

You are an expert in frontend web development and scientific visualization. You are given the title, description, topics, one or more screenshots, and a detailed HTML implementation plan of an interactive scientific demo. Your task is to generate **snapshot-level test cases** that replicate each screenshot through a sequence of realistic UI actions.

### Each test case should:
- Correspond to **one screenshot in the order they appear**.
- Use only **user actions** to reproduce the final state.
- Finish with a screenshot capture of the UI for comparison.

### Test Case Format:
- Title: [Short description of the visual state in screenshot 1/2/3/4]
- Steps:
1. Action: [Simulate the first interaction to reach this state]
2. Action: [Next interaction, if any]
...
N. Action: [Final interaction needed to match the screenshot]
N+1. Assert: Take a screenshot of the current UI state

### Guidelines:
- All interactions must be **derived strictly from the screenshot and the design plan**.
- Test cases must follow the **exact order of input screenshots** (first test case for first screenshot, second test case for second screenshot, etc.).
- If a specific input value is visible (e.g., slider at 0.8), use it.
- If no exact value is visible, describe the interaction in **precise relative terms** (e.g., "drag to 70% of the bar", "click second radio button from left").
- Do **not** speculate about unseen or undocumented behavior.
- The difficulty of each case should be **appropriate**: not trivial (e.g., only opening a page), and not too complex (e.g., involving inferred logic beyond the plan).
- The goal is **accurate UI state replication**, not internal logic testing.

Here is the imformation of this scientific demo and the corresponding implementation plan:
Name: { Name }
Description: { Description }
Topics: { Topics }
Implementation plan:{Implementation plan}
Snapshots: { Snapshots }

### G.4. PFT Unit Test Script Synthesis

**System Prompt - PFT Unit Test Script Synthesis**

You are an expert in frontend web development (HTML, JavaScript, CSS) and scientific visualization. Your task is to generate **Playwright test code** for an interactive HTML page, based on a provided implementation plan and a set of structured **snapshot-level test cases**.

The input includes:
- A detailed **HTML implementation plan**, describing the layout, interactive controls, and how UI state changes based on user input.
- A set of **snapshot-level test cases**, where each test case corresponds to one screenshot and specifies a sequence of user actions to reproduce that state.

### Requirements:
Generate valid Playwright `.spec.js` test code that:
1. Navigates to the local HTML page using the code below.
2. Performs **only the exact user actions** listed in each test case (e.g., drag, click, input).
3. Uses the DOM structure and component IDs specified in the plan—do not guess or infer selectors beyond what is provided.
4. After executing all actions, takes a full-page screenshot of the resulting UI state and saves it as: `await page.screenshot({{path:'./snapshots/{id}-[i].png',fullPage:true}});` where [i] is the index of the test case starting from 1.
5. Ensures that tests are strictly based on the input plan and test cases—do not invent new behaviors or UI logic.
6. Does not rely on function readiness unless stated—only perform initial navigation and DOM load.

### Test Setup:
Load the HTML file using:
`const fileUrl='file://'+require('path').resolve(__dirname,'../pages/{id}.html');}`
No need to wait for external scripts or function readiness beyond page load.

### Output format:
- You must generate only valid complete Playwright test code in JavaScript wrapped in ```javascript and ``` without any explanation.
- Group test cases using `test.describe()` (per group if available).
- Define each test using `test()` with the given test case title.

Here is the implementation plan and test cases:
Implementation plan:{Implementation plan}
Test cases:{Test cases}

### G.5. VQT Unit Test Script Synthesis

**System Prompt - VQT Unit Test Script Synthesis**

You are an expert in frontend web development (HTML, JavaScript, CSS) and scientific visualization. Your task is to generate **Playwright test code** for an interactive HTML page, based on a provided implementation plan and a set of structured **snapshot-level test cases**.

The input includes:
- A detailed **HTML implementation plan**, describing the layout, interactive controls, and how UI state changes based on user input.
- A set of **snapshot-level test cases**, where each test case corresponds to one screenshot and specifies a sequence

of user actions to reproduce that state.

### Requirements:
Generate valid Playwright `.spec.js` test code that:
1. Navigates to the local HTML page using the code below.
2. Performs **only the exact user actions** listed in each test case (e.g., drag, click, input).
3. Uses the DOM structure and component IDs specified in the plan—do not guess or infer selectors beyond what is provided.
4. After executing all actions, takes a full-page screenshot of the resulting UI state and saves it as: `await page.screenshot({{path:'./snapshots/{id}-[i].png',fullPage:true}});` where [i] is the index of the test case starting from 1.
5. Ensures that tests are strictly based on the input plan and test cases—do not invent new behaviors or UI logic.
6. Does not rely on function readiness unless stated—only perform initial navigation and DOM load.

### Test Setup:
Load the HTML file using:
`const fileUrl='file://'+require('path').resolve(__dirname,'../pages/{id}.html');`
No need to wait for external scripts or function readiness beyond page load.

### Output format:
- You must generate only valid complete Playwright test code in JavaScript wrapped in ```javascript and ``` without any explanation.
- Group test cases using `test.describe()` (per group if available).
- Define each test using `test()` with the given test case title.

Here is the implementation plan and test cases:
Implementation plan:{Implementation plan}
Test cases:{Test cases}

## G.6. VQT Checklist Synthesis

**System Prompt - VQT Checklist Synthesis**

You are an expert in frontend web development (HTML, JavaScript, CSS) and scientific visualization.

You are given:
1. A **detailed implementation plan** of an interactive scientific demo, describing the UI structure, control elements (sliders, buttons, dropdowns, text fields), and the theorem or scientific principle the demo explains.
2. A set of **screenshots of the demo under different input states**. Each screenshot shows:
* The **input snapshot** (current state of controls such as slider values, button toggles, dropdown selections).
* The **visual output** (graph, diagram, simulation, or formula rendering) produced by the demo under that input.

Your task is to generate a **checklist for each screenshot**, which specifies the scientific correctness criteria of the **visual output** given that input state.

### Requirements for the checklist
1. **Checklist is output-oriented**
* Do not check whether buttons, sliders, or controls are styled correctly.
* Treat control states only as **inputs** that determine what the visualization should show.
* Focus all checklist items on verifying whether the **visual output image** is scientifically correct.
2. **Connect input to output explicitly**
* Every checklist item must link the given **input state** to the expected **output visualization**.

* Example: *If the angle slider is set to 45°, then the plotted projectile trajectory must peak at the midpoint of its range.*
3. **Scientific focus**
* For math demos: formulas, curves, intersections, asymptotes.
* For physics demos: motion paths, conservation laws, force vectors.
* For geometry demos: shapes, proportions, congruency.
* For statistics/plots: distributions, scaling, labeled values.
4. **Visual verification**
* All checklist items must be verifiable by comparing the **screenshot output image** with a reference screenshot.
* Do not assume hidden internal states or unseen code behavior.
5. **Do not go beyond the plan**
* Only include checklist items that are **explicitly described in the implementation plan** *and* are **visible in the screenshot**.
* Do not invent or infer behavior that is not both in the plan and observable in the screenshot.
* If something is in the plan but not visible in the screenshot, **do not include it**.

### Output Format (strict JSON)

```
{
  "screenshot_id": "n",
  "input_state": {
    "control_name_1": "value",
    "control_name_2": "value"
  },
  "checklist": [
    {
      "category": "Formula correctness",
      "expectation": "{expected formula or update, but only if
      both plan and screenshot show formula}"
    },
    {
      "category": "Graph/Diagram correctness",
      "expectation": "{expected graph or shape properties, if
      plan defines them and screenshot shows them}"
    },
    {
      "category": "Axes/Labels correctness",
      "expectation": "{expected axis labels/ranges/units, only
      if defined in plan and visible}"
    },
    {
      "category": "Numeric outputs correctness",
      "expectation": "{expected numerical outputs, but only if
      plan specifies their presence and screenshot shows them}"
    },
    {
      "category": "Consistency with input state",
      "expectation": "{visual updates must reflect given input
      controls, only if plan links them to visuals}"
    }
  ]
}
```

### Example (from a projectile motion demo plan)

**Screenshot input state**: angle slider = 30°, initial speed = 20 m/s

```
{
  "screenshot_id": "1",
  "input_state": {
    "angle_slider": "30°",
    "initial_speed": "20 m/s"
  },
  "checklist": [
    {
      "category": "Formula correctness",
      "expectation": "Displayed trajectory equation includes theta =
      30° and v = 20 m/s, as defined in the plan"
    },
    {
      "category": "Graph/Diagram correctness",
      "expectation": "Trajectory curve is parabolic as specified in
      plan; arc matches input parameters"
    },
    {
      "category": "Axes/Labels correctness",
      "expectation": "Axes labeled with meters as required by plan"
    },
    {
      "category": "Numeric outputs correctness",
      "expectation": "Maximum height and range values are displayed
      if defined in plan and visible in screenshot"
    },
    {
      "category": "Consistency with input state",
      "expectation": "Visualization reflects input slider angle = 30°"
    }
  ]
}
```
Here is the implementation plan and snapshots:
ID: {ID}
Implementation plan:{Implementation plan}
Snapshots:{Snapshots}

## G.7. VQT VLM-as-Judge

System Prompt - VQT VLM-as-Judge

You are an expert judge for evaluating scientific visualization demos.

You are given:
1. A **reference screenshot** that represents the correct output of the demo under a specific input state.
2. A **generated screenshot** from a candidate implementation under the same input state.
3. A **checklist** of verification items describing what scientific properties must be visible and correct in the output image.

### Your task
1. Carefully compare the **generated screenshot** with the **reference screenshot**.
2. For each **checklist item**, assign a score from **1 to 5** using the rubric below.
3. Provide a short justification for each score.

### Scoring Rubric
* **5 (Perfect / Fully Correct)**
* Output image matches the reference screenshot precisely for this checklist item.
* No scientific or visual errors observed.
* **4 (Minor Deviation)**
* Output image mostly matches, but there are small differences (e.g., slight shift in curve, small scaling error, minor misalignment) that do not change the core scientific correctness.
* **3 (Partial Correctness)**
* Some parts are correct (e.g., correct general shape, but wrong labels; correct axis scaling, but wrong curve peak).
* Noticeable deviation from reference that may reduce scientific clarity.
* **2 (Mostly Incorrect)**
* The item is largely wrong, but a small aspect is still correct (e.g., axis present but mislabeled, trajectory drawn but incorrect path).
* **1 (Completely Incorrect / Missing)**
* The expected scientific property is entirely absent or completely wrong.
* Visualization contradicts the reference screenshot.

### Output Format (strict JSON)

```
{
  "checklist_results": [
    {
      "expectation": "{text from checklist item}",
      "score": 4,
      "reason": "Trajectory shape matches but peak position is
      slightly shifted."
    },
    {
      "expectation": "{text from checklist item}",
      "score": 5,
      "reason": "Axis labels and scaling are identical to reference."
    }
  ]
}
```

Here is the checklist, reference snapshot, and generated snapshot:
Checklist: {Checklist}
Reference snapshot:{Reference snapshot}
Generated snapshot:{Generated snapshot}

## H. discussion

### H.1. Limitations.

Our current test suites are primarily synthesized using Gemini-2.5-Pro. Due to constraints in both domain expertise and annotation costs, we were only able to recruit a small group of graduate students in computer science to validate the generated test scripts and checklists. Their verification combined manual inspection with rule-based checks, allowing us to identify and fix major errors. However, this process cannot guarantee that the entire test suite is free of subtle flaws or omissions, and further large-scale expert validation remains an open challenge.

The benchmark is also sourced from the Wolfram Demonstrations Project, whose demonstrations reflect Mathematica-influenced visual and interaction styles. Our evaluation framework itself is largely style-agnostic because it operates on rendered browser states and DOM behavior, but the current dataset may not cover the full diversity of educational simulation interfaces. Preliminary pilot construction on PhET and GeoGebra demonstrations suggests that the same pipeline can transfer to other UI paradigms, but a systematic cross-platform benchmark remains future work.

Reproducibility is another practical limitation. We use commercial models for high-quality synthesis and VLM judging, so exact regeneration of every artifact requires API access. However, the execution layer of PFT relies only on open-source tools such as Playwright and Node.js, and we release the prompts, data, scripts, and construction pipeline to enable replacement of proprietary components with open-source LLMs or VLMs.

Finally, VQT currently evaluates static screenshots after scripted interactions. This captures many important visual states but cannot fully assess temporal behavior in animations, simulations, or continuous transitions. Future versions should incorporate video-based or trajectory-based visual judging so that temporal consistency, smoothness, and dynamic physical correctness can be evaluated directly. From a societal perspective, benchmarks for scientific demo generation should also be used with care because generated materials may appear plausible while encoding incorrect scientific explanations. We therefore view InteractScience as an evaluation and auditing tool rather than a substitute for expert review in educational deployment.

## H.2. Further Work.

For further work, our evaluation framework is not limited to scientific demonstrations. Given the ease of collecting snapshots, it can be naturally extended to the evaluation of general interactive web applications (Zhang et al., 2025a; Chen et al., 2025). Furthermore, with the rapid progress of GUI-based agents (Qin et al., 2025; Sun et al., 2025b; Liu et al., 2025), agent-driven testing represents a promising future direction that could offer more flexibility and broader applicability than the current scripted approach (Wang et al., 2026; Liang et al., 2024).

Beyond diagnosis, the benchmark can also provide training signals. PFT produces deterministic pass/fail rewards that are directly usable for execution-guided refinement or reinforcement learning, while VQT provides visual and semantic feedback for improving scientific rendering. The cross-discipline results and failure taxonomy identify scientific logic, numerical parameterization, and dynamic visualization as major bottlenecks, offering concrete targets for data collection and model improvement.

