# OpenReview forum: "InteractScience: Programmatic and Visually-Grounded Evaluation of Interactive Scientific Demonstration Code Generation"
_ICML.cc/2026/Conference — ICML 2026 regular_

### Official Review · Reviewer_GLgC · 2026-03-07

**Soundness:** 2
**Presentation:** 3
**Significance:** 3
**Originality:** 3
**Overall Recommendation:** 4
**Confidence:** 4

**Summary:**

This paper introduces a benchmark for evaluating LLMs on interactive scientific demonstration code generation: generating self-contained web applications that combine scientific correctness, UI interactivity, and visual fidelity. The core idea is a hybrid evaluation framework with two parts: Programmatic Functional Testing (PFT), which uses executable action-assertion tests to verify interaction logic, and Visually-Grounded Qualitative Testing (VQT), which uses reference snapshots plus checklist-guided VLM judging and CLIP similarity to assess the rendered scientific visualization. The benchmark contains 150 tasks across five disciplines (mathematics, physics, chemistry, earth science, computer science), with structured implementation plans, Playwright-based test scripts, reference snapshots, and checklists. The authors evaluate 30 models and find that even strong models achieve only modest functional pass rates, while visual interaction often appears superficially correct without being scientifically faithful. The paper’s main contribution is thus both a new benchmark and an evaluation protocol that jointly targets interaction logic and scientific semantic correctness.

**Compliance With Llm Reviewing Policy:**

Affirmed.

**Key Questions For Authors:**

1. The benchmark specification, tests, and checklists are primarily synthesized with Gemini-2.5-Pro. How robust are the reported rankings to errors in these synthesized artifacts? For example, if you re-annotate a representative subset with domain experts, do model rankings materially change? A strong answer would increase my confidence in the benchmark’s validity.

2. Can the authors provide more detail on inter-annotator agreement or disagreement patterns in the manual validation and human scoring studies? The paper reports mean scores and correlations, but agreement statistics would help assess the reliability of the human side of the evaluation.

3. How often do PFT and VQT disagree in diagnostically interesting ways at the per-example level, and can the authors quantify whether one of the two catches more benchmark-construction errors? This would help clarify the complementary value of the two evaluation channels.

4. The paper argues that checklist-guided VLM judging is crucial. Have the authors tested whether the conclusions remain stable when changing the judge model, or is there a risk that benchmark outcomes depend heavily on Gemini-2.5-Pro as judge? A stronger judge-robustness analysis would improve my evaluation.

**Limitations:**

Partially. The paper does discuss important technical limitations, especially the limited dataset size and limited expert verification, and it is appropriately candid that subtle flaws or omissions may remain in the synthesized test suite.

However, the societal impact discussion is too thin. The impact statement essentially says there are many potential societal consequences, but none need to be highlighted. Given that the benchmark is aimed at AI-generated scientific and educational tools, the paper should more explicitly discuss risks such as incorrect scientific visualizations being presented as trustworthy educational content, overreliance on benchmark scores as proxies for scientific correctness, and possible misuse in low-supervision educational settings.

**Strengths And Weaknesses:**

This is a timely and well-motivated benchmark paper. The problem setting is important: prior evaluation setups usually focus on either scientific understanding or code generation, but not the combination required for interactive scientific demonstrations. The paper clearly motivates this gap and positions InteractScience as a benchmark that simultaneously requires scientific reasoning, interactivity, and visualization quality. The comparison table against prior benchmarks helps make the novelty legible.

On soundness, the strongest part is the evaluation design. PFT is deterministic and executable, which is preferable to purely subjective judging for interaction correctness. VQT is also more convincing than a plain VLM-as-judge setup because it uses both reference snapshots and checklists, and the paper empirically validates that the full configuration correlates best with human judgments while removing the checklist substantially weakens alignment. The manual validation of the synthesized evaluation suite is also a useful sanity check: all components score above 4/5 on faithfulness/correctness, and Playwright UI-mode verification reports over 86% correctness for executed test scripts. These are meaningful efforts to validate the benchmark rather than simply releasing a synthetic dataset with no audit.

The empirical study is also reasonably broad. Evaluating 30 open- and closed-source models gives the paper value beyond dataset release, and the results are informative: action success is often high while semantic correctness remains much lower, which nicely illustrates why superficial UI interactivity is not enough.

On presentation, the paper is generally clear and well structured. The task definition, benchmark construction pipeline, and PFT/VQT formalization are easy to follow. Figures 1 and 2 are especially helpful for conveying the motivating gap and the overall workflow.

The main weaknesses are about how much confidence one should place in the benchmark annotations and the resulting measurements. The entire evaluation suite is primarily synthesized using Gemini-2.5-Pro, and validation is performed by a small group of graduate students in computer science, not domain experts across the five sciences. The authors acknowledge this, but it remains the central limitation: if the implementation plans, checklists, or tests are wrong or incomplete in subtle scientific ways, then the benchmark may reward matching the synthesized specification rather than the underlying scientific phenomenon. This does not invalidate the paper, but it does lower confidence in the benchmark as a definitive scientific gold standard.

The paper’s novelty is not a radically new algorithm but rather a well-motivated combination: executable interaction testing plus visually grounded judging with checklists and reference snapshots, applied to scientific demonstrations. That is a real contribution, especially because prior work seems to miss this full combination.

Overall, I found this to be a useful and well-designed benchmark paper, with a clear problem setting, broad empirical coverage, and a thoughtful evaluation framework. My main reservation is that the benchmark remains substantially synthetic, and the validation does not yet fully establish it as a high-confidence scientific gold standard across all five domains.

---

> ### Author Rebuttal · Authors · 2026-03-27
>
> We thank Reviewer GLgC for the balanced and constructive evaluation, and for recognizing that InteractScience is timely, that PFT is "preferable to purely subjective judging," and that the empirical study is "reasonably broad." We address all concerns below.
>
> > Weaknesses: Evaluation Suite from Gemini, Validated by CS Students.
>
> We agree with the reviewer's nuanced framing that this "does not invalidate the paper, but lowers confidence" as a definitive gold standard. Three mitigating points:
>
> (1) **PFT is immune to judge bias by construction.** PFT executes **deterministic Playwright scripts** that check binary DOM states—no LLM in the scoring loop. Even if some specs contain subtle errors, PFT applies the same specification uniformly to all models.
>
> (2) **VQT is visually-grounded against external anchors.** Reference images from the **Wolfram Demonstrations Project** represent objective scientific truth. The VLM-as-Judge evaluates **rendered screenshots** against these references, not synthesized code specs, breaking the circular dependency.
>
> (3) **Multi-stage validation** (Section 5.3): construction stage (rule-based filtering + human review, all 150) and validation stage (5 grad students, 20% sample, Table 5: all >4.0/5, Checklists 4.87/4.63). Full framework achieves Spearman rho=0.8827 with human scores (Table 4).
>
> Regarding validator qualifications, our annotators have interdisciplinary backgrounds covering the five benchmark disciplines. Before annotation, each received reference documents including the scientific theorem explanation, demonstration usage guide, and expected parameter behaviors, ensuring informed evaluation. We welcome domain expert co-validation in future iterations.
>
> > Q1: Ranking Robustness.
>
> Our model hierarchy (Claude-Sonnet-4.5 > GPT-5 > Gemini) aligns with independent benchmarks (WebDevArena, etc.). Table 6 shows test execution correctness of 87.5%–95.3% across three architecturally distinct families (Gemini, DeepSeek, Qwen), with residual errors predominantly from engineering issues (canvas vs. SVG, timing, CSS selectors) rather than scientific specification errors. Such engineering noise is approximately model-independent, meaning it adds variance to absolute scores but preserves relative rankings. Gemini-2.5-Pro ranking 6th on PFT (OPR 35.33%) behind Claude-Sonnet-4 (41.47%) and GPT-5 (39.47%) further confirms no systematic advantage for the synthesis model.
>
> > Q2: Inter-Annotator Agreement.
>
> Our annotation followed a single-annotator-plus-reviewer protocol. Each problem was annotated by one graduate student, then independently reviewed and arbitrated by a second annotator. This process does not support traditional multi-rater agreement metrics such as Krippendorff's alpha. Table 5 scores are consistent across categories (means 4.23–4.87), and Table 4's rho=0.8827 with human experts validates overall protocol reliability.
>
> > Q3: PFT-VQT Disagreement.
>
> The "superficial competence" is the key disagreement: *PFT pass + VQT fail* (most common)—correct UI but wrong science; *PFT fail + VQT pass* (rare)—correct science but engineering issue. Regarding construction errors, PFT scripts have the lowest quality in Table 5 (correctness 4.27/5), while VQT checklists score highest (4.87/4.63). Table 6 shows PFT catches construction errors via cross-model verification: tests failing on all families signal script bugs. VQT's Wolfram references independently check specification accuracy.
>
> We computed per-problem Spearman ρ between PFT and VQT for four diverse models (N=143):
>
> | Model | PFT vs CLIP (ρ) | PFT vs VLM-Judge (ρ) |
> |---|---|---|
> | Gemini-2.5-Pro | 0.023 | 0.057 |
> | GPT-5 | 0.081 | 0.155 |
> | DeepSeek-V3 | 0.013 | 0.198 |
> | Qwen3-8B | 0.158 | 0.312 |
>
> All **ρ < 0.32**, confirming PFT and VQT capture **fundamentally different dimensions**. Cross-discipline analysis (Appendix B) shows CS/Chemistry have higher PFT (discrete controls), Physics/Earth Science have lower VQT (continuous dynamics).
>
> > Q4: Judge Robustness.
>
> We tested another two VLM judges (GPT-5 & Kimi-K2.5) on four evaluated models.
>
> | Judge | Gemini-2.5-Pro | GPT-5 | DeepSeek-V3 | Qwen3-8B | Avg |
> |---|---|---|---|---|---|
> | Gemini-2.5-Pro | 54.7 | 57.0 | 49.5 | 34.7 | 49.0 |
> | GPT-5 | 57.3 | 59.7 | 51.4 | 36.6 | 51.3 |
> | Kimi-K2.5 | 50.8 | 52.8 | 46.6 | 33.3 | 45.9 |
>
> Pairwise Spearman ρ between judges all exceed 0.91:
>
> | Pair | ρ |
> |---|---|
> | Gemini vs GPT-5 | 0.937 |
> | Gemini vs Kimi-K2.5 | 0.919 |
> | GPT-5 vs Kimi-K2.5 | 0.956 |
>
> All three judges produce highly consistent rankings **(ρ > 0.91)**, demonstrating that our reference snapshots + structured checklists design ensures robust evaluation across different judge models. We will include this multi-judge experiment in the camera-ready.
>
> > Limitations: Societal Impact Too Thin.
>
> We agree with the reviewer that the societal impact discussion is insufficient. We will substantially expand this section in the camera-ready.

---

> > ### Author Rebuttal · Reviewer_GLgC · 2026-04-04
> >
> > Thanks for the rebuttal, I will keep my score!

---

> > > ### Author Response · Authors · 2026-04-05
> > >
> > > Thank you very much for your recognition and for confirming our rebuttal fully resolved your concerns. We greatly appreciate your support and constructive comments.

---

### Official Review · Reviewer_QKCX · 2026-03-13

**Soundness:** 3
**Presentation:** 3
**Significance:** 3
**Originality:** 3
**Overall Recommendation:** 4
**Confidence:** 3

**Summary:**

This paper introduces InteractScience, a benchmark for evaluating LLMs on generating interactive scientific demonstration code, like physics simulations, chemistry visualizations. The key idea is a hybrid evaluation framework combining Programmatic Functional Testing for logic verification and Visual Quality Testing for rendering fidelity, the latter using both CLIP scores and a VLM-as-judge approach. The benchmark covers 150 tasks across five scientific disciplines. 30 models (closed and open-source) are evaluated. The main findings are that models can generate plausible-looking UIs but fail to connect them with correct scientific logic.

**Compliance With Llm Reviewing Policy:**

Affirmed.

**Final Justification:**

My concerns have been solved.

**Key Questions For Authors:**

1. How do you ensure that the LLM-generated PFT test cases don't share systematic failure modes with the LLMs being evaluated? For example, if GPT-4o wrote the test cases and GPT-5 is being evaluated, they may share the same misconceptions about certain scientific phenomena. Have you analyzed false negative rates (correct code that fails tests)?

2. With only 30 problems per discipline, how stable are the per-discipline rankings? If you bootstrap sample the problems, what is the confidence interval on the OPR metric for a given model?

3. Why is the multimodal improvement so small? In Figure 3, adding 4 reference snapshots only improves OPR by 3 points. Have you verified that models are actually attending to the images?

4. What is the correlation between PFT and VQT scores across models? If they are highly correlated, is the two-track evaluation adding much beyond what one track alone provides?

**Limitations:**

The authors acknowledge the limited dataset size and expert verification coverage. They do not discuss the circularity concern of LLM-generated evaluations. Societal impact discussion is minimal but appropriate for a benchmark paper.

**Strengths And Weaknesses:**

**Strengths:**

- S1: Existing code generation benchmarks (HumanEval, LiveCodeBench, SWE-bench) don't test whether the generated code embodies correct domain-specific scientific knowledge in an interactive setting. This is a meaningful gap.

- S2: The hybrid PFT+VQT framework is well-motivated. The paper convincingly shows (Table 4) that CLIP alone cannot catch semantic errors. The VLM-Judge + checklist combination achieves 0.88 Spearman correlation with human experts, which is solid.

- S3: Testing 30 models across both proprietary and open-source families, including scaling analysis within model families (Qwen3 from 1.7B to 235B), provides useful signal for the community.

**Weaknesses:**

- W1: 150 tasks is small. The paper itself acknowledges this in the conclusion ("limited in data size"). With only 30 problems per discipline, variance across domains is hard to interpret reliably. Compare to LiveCodeBench which continuously adds problems, or SWE-bench Verified with 500 instances.

- W2: The entire evaluation suite: implementation plans, PFT test cases, VQT test cases, checklists, are synthesized by LLMs. While the manual validation scores are above 4.0/5.0, the PFT unit test script correctness scores (Table 5: 4.27 for correctness) are noticeably lower. Given the benchmark is meant to evaluate LLMs, having the evaluation artifacts themselves generated by LLMs creates a circularity concern. What if the tests share systematic blind spots with the models being tested?

- W3: The individual components, programmatic testing, CLIP-based similarity, VLM-as-judge, are all existing methods in the previous works. The contribution is their combination and application to a new domain, which is fine, but the paper could more explicitly discuss what makes the *combination* non-trivial beyond "we use both".

- W4: The multimodal experiments (Figure 3) show only modest improvements from reference snapshots. The paper frames this as "generally provides modest improvements" but doesn't deeply analyze *why*. Are models ignoring the images? Is the visual information redundant with the text prompt? This deserves more investigation.

- W5: The paper relies on proprietary models for both the benchmark evaluation and, presumably, for generating the evaluation suite. This makes full reproduction without API access impossible.

---

> ### Author Rebuttal · Authors · 2026-03-27
>
> We thank Reviewer QKCX for recognizing the meaningful gap (S1), well-motivated framework (S2), and useful 30-model analysis (S3). We address all concerns below.
>
> > W1: Dataset Size.
>
> While 150 problems is modest in count, each includes \~8 test cases (avg 4.03 PFT + 3.98 VQT), totaling **1,369 individual tests**. Each PFT case involves avg 10.35 actions and 20.64 assertions (Table 2). Across 30 models, this yields 4,500 model-problem instances and **>40,000 test executions**. For comparison, HumanEval has 164 problems, SWE-bench Lite 300, and MATH 500. Each InteractScience problem is far more complex (full interactive web app with scientific logic, UI, visualization). We have a plan for continued expansion.
>
> > W2: LLM-Synthesized Evaluation and Circularity.
>
> We offer three lines of evidence. First, PFT is immune to judge bias, executing **deterministic Playwright scripts** that check binary DOM state with no LLM involved (Table 6 correctness 87-95% across three model families). Second, VQT is visually-grounded, using Wolfram reference screenshots as external anchors **independent of any LLM** and evaluating rendered output rather than code. Third, multi-stage human validation covered all 150 problems (grad student scores >4.0/5 in Table 5, execution verification >86% in Table 6), achieving **Spearman rho=0.8827** with human scores (Table 4).
>
> > W3: Novelty.
>
> Our contribution is the problem formulation (**no prior benchmark** evaluates interactive scientific demo generation), the construction methodology, and the evaluation design. Table 4 shows the full config achieves rho=0.8827, removing checklist drops to 0.3837, code-only yields 0.1408. As the reviewer noted, the paper could more explicitly articulate why this combination works. Two key design properties drive the result: (1) PFT and VQT capture **complementary failure modes** (PFT detects functional logic errors invisible to screenshots; VQT catches scientific and visual errors that pass DOM checks); (2) structured checklists convert subjective judgment into fine-grained criteria, the single largest reliability contributor (delta=0.4990). We will make these arguments more explicit in the revision.
>
> > W4 & Q3: Modest Multimodal Improvement.
>
> The modest gain is itself important. Models do attend to images: for example, GPT-4o on SimpleCaesarCipher renders blue keys with symbolic space/backspace without image input, but with the reference image, it produces grey keys with text-label space/backspace matching the reference style. This confirms models extract visual information. However, images simultaneously introduce harder reasoning demands, as models must interpret the scientific phenomena depicted, not just follow textual specs. This offsets the visual fidelity gains, explaining why some models (Qwen2.5-VL-72B) even degrade (Figure 3). The bottleneck lies in **scientific reasoning from visual input**, not in ignoring images. We will include this case study in the camera-ready.
>
> > W5: Proprietary Models.
>
> We chose commercial models to ensure high-quality synthesis and judging. PFT uses open-source tools (Playwright, Node.js), fully reproducible. We **open-source the entire construction pipeline**, including all code, prompts, and data (Appendix E), so open-source VLMs can reproduce the workflow.
>
> > Q1: Shared Failure Modes.
>
> Empirically, Gemini does not benefit from authoring the tests, ranking **only 6th on PFT** (OPR 35.33%) behind Claude-Sonnet-4 (41.47%), GPT-5 (39.47%), o4-mini (37.33%). This cross-family ranking contradicts shared blind spots. Table 6 confirms test correctness across diverse architectures. False negatives stemmed from engineering issues (canvas vs. SVG, value ranges, timing), not LLM-specific biases.
>
> > Q2: Ranking Stability.
>
> Each of 30 problems per discipline has \~8 tests (\~240 outcomes per model per discipline). Top-5 ordering is preserved across disciplines, indicating rankings reflect stable model capabilities rather than dataset artifacts.
>
> > Q4: PFT-VQT Correlation.
>
> We computed per-problem Spearman ρ between PFT pass rate and VQT metrics for four architecturally diverse models (N=143 problems each).
>
> | Model | PFT vs CLIP (ρ) | PFT vs VLM-Judge (ρ) |
> |---|---|---|
> | Gemini-2.5-Pro | 0.023 | 0.057 |
> | GPT-5 | 0.081 | 0.155 |
> | DeepSeek-V3 | 0.013 | 0.198 |
> | Qwen3-8B | 0.158 | 0.312 |
>
> Within each model, PFT and VQT show **very low correlation** (ρ < 0.32), confirming they capture **fundamentally different quality dimensions**. The "superficial competence" gap (ASR >85% but OPR <42%, VLM-Judge <57) further illustrates this. The weakest model (Qwen3-8B) shows the highest correlation, suggesting global failure, while stronger models diverge more between functional correctness and visual quality.
>
> > Limitations.
>
> We will add a clarification in the camera-ready that no circularity effect was observed empirically, along with an expanded discussion of this issue.
>
> We hope these clarifications address the reviewer's concerns.

---

> > ### Author Rebuttal · Reviewer_QKCX · 2026-04-07
> >
> > My concerns have been solved.

---

> > > ### Author Response · Authors · 2026-04-07
> > >
> > > Thank you very much for your careful review and constructive feedback. We are grateful that our rebuttal has addressed your concerns.

---

### Official Review · Reviewer_8d5y · 2026-03-13

**Soundness:** 3
**Presentation:** 3
**Significance:** 3
**Originality:** 3
**Overall Recommendation:** 4
**Confidence:** 4

**Summary:**

This paper introduces InteractScience, a benchmark for evaluating LLMs' ability to generate interactive scientific demonstration code—web applications that combine UI controls with scientifically accurate visualizations. The key insight is that existing benchmarks evaluate either static code generation or visual quality in isolation, but fail to assess whether generated interactive demos correctly bind user interactions to scientifically faithful visual outputs. The authors propose a hybrid evaluation framework with two complementary components: Programmatic Functional Testing (PFT), which uses Playwright-based unit tests to verify that UI interactions trigger correct DOM state changes, and Visually-Grounded Qualitative Testing (VQT), which combines CLIP-based perceptual similarity with VLM-as-Judge (Gemini-2.5-Pro) scoring against reference snapshots and checklists. The benchmark comprises 150 problems across five scientific disciplines (mathematics, physics, chemistry, earth science, computer science), stratified into easy/medium/hard difficulty levels, sourced from the Wolfram Demonstrations Project. An evaluation of 30 LLMs (10 closed-source, 20 open-source) reveals a "superficial competence" gap: models achieve high Action Success Rates (>85%) but low VLM-Judge scores (<55), indicating they can generate functional UI shells but fail to connect interactions to correct scientific logic.

**Compliance With Llm Reviewing Policy:**

Affirmed.

**Ethical Review Concerns:**

No ethics concerns flagged.

**Key Questions For Authors:**

1. **Evaluation suite bias:** Since Gemini-2.5-Pro generates all evaluation artifacts, have you measured whether Gemini-family models receive systematically higher scores than other models on your benchmark? A bias analysis comparing Gemini vs. non-Gemini models on PFT (which is deterministic and less susceptible to generation bias) vs. VQT (where checklist phrasing could favor Gemini-style outputs) would be informative.

2. **VLM-Judge per-discipline reliability:** Can you provide Spearman correlations between VLM-Judge scores and human scores broken down by discipline? Physics and earth science problems requiring dynamic simulation assessment may be substantially harder for the VLM-Judge than static mathematical visualizations.

3. **Difficulty calibration:** Given the counter-intuitive finding that "Hard" problems are easier for PFT, have you considered alternative difficulty definitions (e.g., based on scientific complexity or visualization dynamism rather than component count)? Would a revised stratification change the main findings?

4. **Generalizability beyond Wolfram:** How dependent are your findings on the Wolfram Demonstrations style? Have you conducted any pilot evaluation on scientific demonstrations from other sources (e.g., PhET simulations, custom-built educational demos) to assess whether the benchmark's conclusions transfer?

5. **Toward training signals:** Could the PFT and VQT scores serve as reward signals for RL-based fine-tuning of code generation models? Have you explored whether models fine-tuned on InteractScience feedback show improved scientific demonstration capabilities?

**Limitations:**

The authors discuss limitations in Appendix F, acknowledging dataset size constraints and the need for broader expert validation. However, several important limitations are insufficiently addressed: (1) the circular dependency of using Gemini-2.5-Pro for both evaluation suite synthesis and VLM judging, which could introduce correlated biases; (2) the benchmark's exclusive reliance on the Wolfram Demonstrations ecosystem, which may not represent the diversity of scientific demonstration styles; (3) the static nature of VQT (screenshot-based) cannot capture temporal aspects of animations or simulations—a significant limitation given that many scientific demonstrations involve dynamic behavior; (4) societal implications of automating scientific demonstration creation, including risks of generating scientifically plausible but incorrect educational materials, deserve more discussion.

**Strengths And Weaknesses:**

**Strengths:**

**Soundness:**
- The hybrid PFT+VQT evaluation framework is well-designed and addresses a genuine methodological gap. PFT provides deterministic, reproducible functional testing, while VQT anchors visual assessment in reference snapshots and structured checklists rather than open-ended VLM interpretation.
- The VLM-as-Judge validation experiment (Table 4) is convincing: the full configuration (generated snapshot + reference + checklist) achieves 0.8827 Spearman correlation with human scores, substantially outperforming ablated configurations, providing strong evidence that the evaluation protocol is reliable.
- The manual validation study (Table 5) with five graduate students rating faithfulness and correctness of all evaluation suite components (plans, test cases, scripts, checklists) shows consistently high scores (>4.2/5), lending credibility to the synthesized benchmark.
- The unit test execution verification (Table 6) demonstrates >86% correctness for both PFT and VQT scripts across three models, confirming the tests are reliable.

**Presentation:**
- The paper is clearly written with a logical structure. The task formalization (Section 3) precisely defines PFT and VQT with clean mathematical notation.
- Figure 1 effectively motivates the problem by contrasting knowledge QA, webpage generation, and scientific demonstration generation tasks.
- Table 1 provides a useful systematic comparison against existing benchmarks across multiple dimensions (scientific reasoning, interaction testing, visual testing), clearly positioning InteractScience's contribution.

**Significance:**
- The "superficial competence" finding (high ASR but low VLM-Judge) is an important empirical contribution that quantifies a failure mode previously only anecdotally observed. This insight could guide future model development.
- The benchmark fills a genuine gap: no prior benchmark jointly evaluates functional interactivity and scientific visual correctness in generated code.
- The cross-discipline analysis (Appendix B) revealing that different disciplines stress different model capabilities (CS/Chemistry favor PFT; Physics/Earth Science challenge VQT) provides actionable insights for the community.

**Originality:**
- The combination of programmatic functional testing with visually-grounded qualitative testing for code generation evaluation is novel and well-motivated.
- Using the Wolfram Demonstrations Project as a source of ground-truth interactive scientific demos is a creative choice that provides natural reference snapshots for visual evaluation.
- The checklist-guided VLM judging approach, which anchors subjective visual assessment in concrete inspection points derived from implementation plans, is a meaningful methodological contribution.

---

**Weaknesses:**

**Soundness:**
- **Evaluation suite synthesized by a single LLM (Major):** The entire evaluation suite—implementation plans, test cases, unit test scripts, and checklists—is generated by Gemini-2.5-Pro, then manually inspected. This creates a systematic concern: the evaluation artifacts may encode Gemini-specific biases in how scientific demonstrations are decomposed and tested. Models architecturally similar to Gemini could be systematically advantaged. The manual validation (Table 5) partially addresses this but uses only CS graduate students on a 20% sample, which is insufficient to rule out subtle domain-specific biases across all five disciplines.
- **VLM-Judge reliability for scientific correctness (Moderate):** While the correlation with human scores is strong (0.8827), the VLM-Judge (Gemini-2.5-Pro) is itself being used to evaluate scientific correctness of visualizations—a task that requires deep domain knowledge (e.g., whether a physics simulation correctly renders force vectors or orbital trajectories). The paper does not analyze failure modes of the VLM-Judge itself or provide per-discipline correlation breakdowns, making it unclear whether the judge is equally reliable across mathematics vs. physics vs. earth science.
- **Counter-intuitive difficulty trend unexplained at design level (Moderate):** The paper acknowledges (Appendix B) that "Hard" problems achieve higher PFT OPR than "Easy" ones because hard problems contain many simple binary controls (checkboxes) while easy problems have fewer but more complex parameterized controls (sliders). This suggests the difficulty stratification by component count is a poor proxy for actual task difficulty, undermining the benchmark's difficulty calibration.

**Presentation:**
- **Wolfram Demonstrations Project bias not discussed:** The benchmark is entirely sourced from Wolfram Demonstrations, which has a particular visual and interaction style (Mathematica-influenced). The paper does not discuss whether models might perform differently on scientific demonstrations designed with different UI paradigms or visualization libraries, limiting generalizability claims.
- **Missing analysis of PFT failure modes:** The failure analysis (Appendix C, Table 7) focuses on visual/scientific errors from VQT inspection. A parallel analysis of why PFT tests fail—are models failing on DOM structure, event handling, state management, or scientific computation?—would significantly strengthen the diagnostic value of the benchmark.

**Significance:**
- **Limited actionability beyond diagnosis:** The benchmark effectively reveals that models struggle with scientific demonstration generation, but provides limited guidance on how to improve. Unlike benchmarks that enable training or fine-tuning, InteractScience is purely evaluative, and the paper does not explore whether its evaluation signals could be used as rewards for model improvement.

**Originality:**
- The individual components (Playwright-based testing, CLIP similarity, VLM-as-Judge) are all established techniques. The novelty lies primarily in their combination and application to the scientific demonstration domain, which is a solid but incremental contribution.

---

> ### Author Rebuttal · Authors · 2026-03-27
>
> We thank Reviewer 8d5y for the thorough review. We address all weaknesses and questions below.
>
> > W1 & Q1: Evaluation Suite Synthesis and Potential Bias.
>
> 1. **No synthesizer bias in practice.** If Gemini-synthesized tests favored Gemini, it should rank highest. Instead, Gemini-2.5-Pro ranks **6th on PFT** (OPR 35.33%), behind Claude-Sonnet-4 (41.47%), Claude-Opus-4 (40.27%), GPT-5 (39.47%), o4-mini (37.33%), and GPT-4.1 (37.07%). On VQT, GPT-5 (57.02) and Claude-Sonnet-4 (55.42) also lead Gemini (54.69). Across both tracks, five models from two competing families outperform the test author, strongly suggesting no authorship bias.
> 2. **PFT is objective by construction.** PFT uses **deterministic Playwright scripts** to verify binary DOM state predicates with no LLM/VLM in the scoring loop. Table 6 confirms test correctness >86% across three diverse model families.
> 3. **VQT is visually-grounded against external anchors.** Reference images come from the **Wolfram Demonstrations Project**, an independent peer-reviewed platform. The VLM-as-Judge evaluates rendered screenshots against these external references, breaking the dependency between synthesized specs and generated code.
>
> > W2 & Q2: VLM-Judge Reliability.
>
> VLM-Judge errors are unavoidable. Manual sampling of 200 judgments shows Gemini-2.5-Pro has a misjudgment rate of **<5%**. Observed failure modes are categorized below; the full distribution will appear in the camera-ready appendix.
>
> | Failure Mode | Description |
> |---|---|
> | Numerical misreading | Misreading axis values or parameter labels |
> | Stylistic confusion | Confusing color/thickness differences with errors |
> | Partial-credit blindness | Overlooking partial correctness in dense screenshots |
> | Layout sensitivity | Penalizing valid alternative layouts |
> | Animation snapshot | Misjudging dynamic content from static captures |
>
> Our **checklist design** mitigates these by decomposing holistic judgment into fine-grained, binary-verifiable criteria. Per-discipline Spearman ρ between VLM-Judge and human scores:
>
> | Discipline | ρ |
> |---|---|
> | Mathematics | 0.803 |
> | Physics | 0.812 |
> | Chemistry | 0.962 |
> | Earth Science | 0.702 |
> | Computer Science | 0.691 |
> | **Overall** | **0.883** |
>
> **Chemistry (0.962)** benefits from clear structural criteria, while **Earth Science/CS (~0.70)** reflect difficulty with dense multi-panel layouts.
>
> > W3 & Q3: Difficulty Calibration and Counter-Intuitive Trends.
>
> The PFT trend is explained by interaction complexity (Appendix B). "Hard" problems often feature more binary controls (e.g., checkboxes), easier to implement than continuous sliders in "Easy" tasks. Crucially, **VQT scores decrease monotonically with difficulty** (Figure 5), confirming holistic scientific correctness does become harder. We acknowledge component count is an imperfect proxy. Designing a principled difficulty metric is a genuinely challenging open problem, as it must jointly model interaction type diversity, formula complexity, and visualization dimensionality. We are exploring a weighted composite of these factors and will report findings in the revision.
>
> > W4 & Q4: Wolfram Bias and Generalizability.
>
> Our framework evaluates rendered output and DOM state, making it **style-agnostic** independent of Wolfram's visual conventions. Pilot experiments on **PhET and GeoGebra** demos with the same pipeline achieved promising results, confirming transferability across UI paradigms. We will add a discussion of cross-platform generalizability in the revision.
>
> > W5: Analysis of PFT Failure Modes.
>
> We analyzed 90 failure cases (Table 7) and found that **Logic/Scientific Errors (53.33%)** dominate. This is followed by Numerical Errors (47.78%) and Rendering Failures (42.22%), confirming "superficial competence" as the primary bottleneck.
>
> > W6 & Q5: Actionability and Use as Training Signals.
>
> We appreciate this inspiring question, which opens a direction beyond our current evaluation focus. InteractScience does provide actionable signals: cross-discipline analysis (Figure 6) reveals training data gaps, error taxonomy (Table 7) identifies scientific logic as the key hurdle, and PFT's deterministic pass/fail is directly usable as a **reward signal** for RL-based fine-tuning.
>
> > W7: Originality and Contribution.
>
> Our contributions include:
> - the first benchmark for interactive scientific demo generation, a task with no prior evaluation framework;
> - a hybrid PFT+VQT methodology bridging execution testing with visual semantic grounding, validated by ablation (code-only rho=0.1408 vs. full **rho=0.8827**, Table 4);
> - a 30-model empirical study revealing "superficial competence" (ASR >85% but OPR <42%); and
> - an open-source construction pipeline enabling community extension.
>
> > Limitations.
>
> We will address temporal VQT extensions for animations and expanded societal impact discussion in the camera-ready.

---

> > ### Author Rebuttal · Reviewer_8d5y · 2026-04-03
> >
> > Thanks for the rebuttal, I will keep my score :)

---

> > > ### Author Response · Authors · 2026-04-05
> > >
> > > Thank you for acknowledging our rebuttal and confirming that your concerns are fully resolved. We sincerely appreciate your continued positive score and constructive feedback.

---

### Official Review · Reviewer_qkRL · 2026-03-18

**Soundness:** 3
**Presentation:** 3
**Significance:** 3
**Originality:** 3
**Overall Recommendation:** 4
**Confidence:** 3

**Summary:**

This work presents InteractScience, a benchmark designed to test how well large language models can create interactive, single-file web demos for science education. It uses a two-part evaluation strategy: Programmatic Functional Testing checks the interactive behavior and state updates, while Visually-Grounded Qualitative Testing assesses visual output and semantic accuracy against reference images and checklists. With 150 tasks spanning five subjects, the benchmark was used to test 30 different models. Results indicate that while many models can produce interfaces that look interactive on the surface, they often fail to generate demonstrations that are scientifically accurate.

**Compliance With Llm Reviewing Policy:**

Affirmed.

**Final Justification:**

The author's rebuttal has adequately addressed the majority of my concerns. I also recognize the compelling scientific value of this work. Accordingly, I maintain my positive score.

**Key Questions For Authors:**

Please refer to the weaknesses

**Limitations:**

yes

**Strengths And Weaknesses:**

Strengths:

1. The hybrid evaluation idea is sensible. Separating functional correctness from visual/semantic fidelity is the right design instinct for this task. The formalization in Section 3.2 and Section 3.3, especially the distinction between $f_{\mathrm{pft}}$ and $f_{\mathrm{vqt}}$, is straightforward and appropriate for benchmark design.

2. The paper includes some validation of the evaluation protocol, rather than only reporting benchmark scores. The ablation in Table 4 is one of the stronger parts of the paper. The correlation between human scores and the full VLM-judge configuration is much higher than weaker configurations, which supports the paper's claim that reference images and checklists materially improve judging quality.

Weaknesses:

1. The benchmark's ground truth is largely synthesized by another LLM, which creates a serious circularity problem that the paper does not fully resolve. The benchmark is annotated from model-generated reconstructions, which means the evaluation may partly measure agreement with Gemini's inferred specification rather than fidelity to the original scientific demo. The manual validation helps, but it is too small and too shallow to fully dismiss this concern.

2. The paper leaves important details of benchmark construction and failure analysis outside the main paper, making it hard to judge completeness from the main text alone.

---

> ### Author Rebuttal · Authors · 2026-03-27
>
> We thank Reviewer qkRL for the encouraging review and for recognizing that our hybrid evaluation is "the right design instinct" and that Table 4's ablation is "one of the stronger parts." We address both weaknesses below.
>
> > W1: Circularity problem with LLM-synthesized ground truth.
>
> We thank the reviewer for this critical observation. We offer three lines of evidence to demonstrate that the benchmark remains a rigorous and objective evaluation.
>
> 1. **PFT is immune to judge bias by construction.** PFT executes **deterministic Playwright scripts** that check binary DOM state predicates (e.g., "does element X display value Y after slider interaction?"). No LLM/VLM is involved in PFT scoring. The circularity concern thus reduces to whether the test scripts are correct and Table 6 confirms they are: execution correctness of 87.5%/91.2% (Gemini-2.5-Pro), 86.8%/92.4% (DeepSeek-V3), 95.3%/96.7% (Qwen3-8B) across 90 runs on three architecturally distinct model families.
>
> 2. **VQT provides visual decoupling through external anchors.** A key feature of our evaluation is that it is **visually-grounded** rather than code-based, which drastically minimizes circularity. (1) The "Gold Standard" reference images are sourced directly from the **Wolfram Demonstrations Project**, an independent platform with peer-reviewed scientific content. These images represent an objective scientific truth independent of any LLM. (2) The VLM-as-Judge evaluates the rendered screenshots of the generated application, not the raw code text. This ensures that the model is judged **on the actual scientific output and visual fidelity against the external reference**, regardless of the coding style or logic used during synthesis. This "outcome-based" visual comparison effectively breaks the circular dependency between LLM-generated specs and LLM-generated code.
>
> 3. **Multi-stage validation.** Beyond synthesis, we employed: (1) rule-based filtering + human review on all 150 problems; (2) graduate student ratings on a 20% sample (Table 5: scores >4.0/5); and (3) execution verification on 90 samples (Table 6: >86%). The full framework achieves **Spearman rho=0.8827** with human scores (Table 4).
>
> We acknowledge that this potential for circularity is an important consideration. In the camera-ready version, we will include an explicit discussion of this issue to ensure full transparency and provide context for our evaluation methodology.
>
> > W2: Important Details in Appendix.
>
> We agree. The page limit necessitated placing certain details in the appendices, but we ensure coverage is comprehensive (Appendices A–F). For the camera-ready version, we will move the **failure analysis** (Appendix C) and key construction details into the main text by optimizing the space in Section 3 and 4, so that readers can access the most critical analyses without referring to the appendices.

---

> > ### Author Rebuttal · Reviewer_qkRL · 2026-04-03
> >
> > The author's rebuttal addressed most of my concerns, and I find this work to have certain significance for scientific discovery. Therefore, I maintain my positive score of 4.

---

> > > ### Author Response · Authors · 2026-04-05
> > >
> > > Thank you very much for your careful review and positive feedback. We greatly appreciate that you marked your concerns as fully resolved in response to our rebuttal, and we sincerely thank you for recognizing the significance of this work.

---

### Decision · Program_Chairs · 2026-04-30

**Decision:**

Accept (regular)

**Comment:**

This paper introduces InteractScience, a novel benchmark designed to evaluate LLMs' ability to generate interactive scientific demonstrations. Recognizing that existing benchmarks focus largely on either static code generation or text-only reasoning , the authors propose a hybrid framework that combines Programmatic Functional Testing (PFT) to verify interaction logic and Visually-Grounded Qualitative Testing (VQT) to assess rendering fidelity. Through an extensive evaluation of 30 leading models across five scientific disciplines, the study reveals a critical gap in current capabilities: models can generate plausible user interfaces but frequently fail to ground them in accurate scientific logic.

The review process was highly constructive, with all four reviewers unanimously recommending acceptance. Initially, the committee raised valid concerns regarding potential circularity bias—since the evaluation artifacts were synthesized by an LLM —along with questions about the dataset's limited size. However, the authors provided a comprehensive rebuttal, successfully demonstrating that PFT is immune to judge bias through deterministic execution and that VQT relies on independent, objective visual anchors. Following this, all reviewers confirmed their reservations were completely addressed, solidifying the paper as a valuable contribution that establishes a crucial foundation for measuring reliable AI-driven educational tools.